# Cross-Quality Few-Shot Transfer for Alloy Yield Strength Prediction: A New Material Science Benchmark and An Integrated Optimization Framework

## Abstract

Discovering high-entropy alloys (HEAs) with high yield strength is an important yet challenging task in material science. However, the yield strength can only be accurately measured by very expensive and time-consuming real-world experiments, hence cannot be acquired at scale. Learning-based methods could facilitate the discovery process, but the lack of a comprehensive dataset on HEA yield strength has created barriers. We present **X-Yield**, a large-scale material science benchmark with 240 experimentally measured ("high-quality") and over 100K simulated (imperfect or "low-quality") HEA yield strength annotations. Due to the scarcity of experimental annotations and the quality gap in imperfectly simulated data, existing transfer learning methods cannot generalize well on our dataset. We address this *cross-quality few-shot transfer* problem by leveraging model sparsification "twice" — as a noise-robust feature learning regularizer at the pre-training stage, and as a data-efficient learning regularizer at the few-shot transfer stage. While the workflow already performs decently with ad-hoc sparsity patterns tuned independently for either stage, we take a step further by proposing a bi-level optimization framework termed **Bi-RPT**, that jointly learns optimal masks and automatically allocates sparsity levels for both stages. The optimization problem is solved efficiently using gradient unrolling, which is seamlessly integrated with the training process. The effectiveness of Bi-RPT is validated through extensive experiments on our new challenging X-Yield dataset, alongside other synthesized testbeds. Specifically, we achieve an $8.9 \sim 19.8\%$ reduction in terms of the test mean squared error and $0.98 \sim 1.53\%$ in terms of test accuracy, merely using 5-10% of the experimental data. Codes and sample data are in the supplement.

## 1 Introduction

Machine learning (ML) methods have recently demonstrated great promise in the important field of material science, and in this paper, we focus on ML-assisted high-entropy alloy (HEA) (Yeh et al., 2004) discovery and property prediction. HEAs own promising properties that traditional alloys do not hold, such as extraordinary mechanical performance at high temperatures, making them well-suited options for various material applications. One particular property, *i.e.,* the yield strength of HEAs, characterizes the maximum stress a material can endure before starting to deform, which is a critical parameter for customized HEA design.

However, in order to accurately measure the yield strength of specific HEAs, expensive scientific experiments need to be conducted for each alloy, often involving hard-to-create experimental conditions, especially at high temperatures (mainly caused by difficulties with oxidation control) as well as extremely long experimental duration. At high temperatures, these measurements are typically taken with the Gleeble system (Gle). From sample preparation to yield strength measurement can take between two to four weeks even for a domain expert team, including melting of the alloy, machining the sample, and preparing and mechanically testing with the Gleeble. Therefore, it is challenging to acquire yield strength measurements from those "high-quality" experiments at scale.

Similar to the trends in computer vision fields (Tremblay et al., 2018), recent efforts attempt to mitigate the scarcity of real-world measurements using ML-based predictors: to directly predict their yield strengths from the alloy inputs (Bhandari et al., 2021a); and such ML-based predictors could be trained using simulated data. Indeed, material sciences applications are often blessed by developed simulation models, e.g., Maresca & Curtin (2020). However, such a blessing is often compromised by the domain gap between the simulated data and the "ground-truth" experimental data, often due to many inevitable simplifications in simulation modeling. For example, the yield strength of a material can vary greatly based on processing and testing conditions as well as grain size and texture (Toda-Carballo et al., 2014; Lin et al., 2014); yet simulation models commonly rely on properties intrinsic to the alloy and do not incorporate variations in experimental conditions. The lack of public datasets in this field also renders it difficult to benchmark ML models' progress.

In this paper, we start by curating a large-scale benchmark, called **X-Yield**, that for the first time combines experimental data with simulation data to address the problem of predicting yield strength in HEAs. While using experimental data is always preferred since they are *"high-quality"* ground truths, it is impractical to generate high quantities of data, especially for capturing yield strength at elevated temperatures. Thus, simulation data can be acquired by massive quantities to fill the gap, despite their relatively *"low quality"* due to inherent model misspecification or simplification. The low-quality simulation data was selected to represent ternary-septenary systems from an eleven-element palette consisting of mostly refractory elements (Al-Cr-Fe-Hf-Mo-Nb-Ta-Ti-V-W-Zr). While there are existing experimental databases (Borg et al., 2020) and models to predict high-temperature yield strength in HEAs (Maresca & Curtin, 2020), to our best knowledge, this is **the first multi-fidelity dataset in the public domain** that combines real experimental measurements and large quantities (over 100K) of simulation data for mechanical property prediction in HEAs. This specialized data set should be able to predict high-temperature yield strength across a broad range of HEAs. The predictions of this model could be used to pinpoint which alloys are the strongest at elevated temperatures, allowing experiments to focus on pre-sorted candidates for future study eliminating the need to spend several weeks testing a candidate without promise.

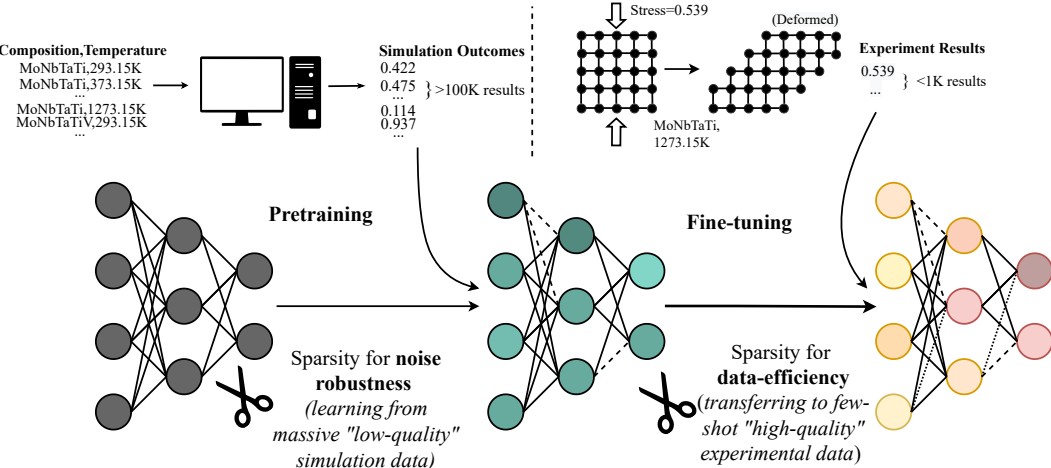

Figure 1: Proposed two-stage workflow. The HEA yield strength prediction model is first pre-trained on massive "low-quality" simulation data, and is then fine-tuned/transferred on few-shot "high-quality" experimental data to optimize its prediction in this target domain. Note that the tool of sparsity will be leveraged in both pre-training and fine-tuning stages, for the purposes of gaining noise robustness/transferablity and enhancing data efficiency, respectively.

The new X-Yield benchmark is set to facilitate ML for HEA yield strength prediction, but learning from such a multi-fidelity dataset is highly non-trivial. To this end, we next conceptualize a **cross-quality few-shot transfer workflow**: first pre-training the prediction model on the data-rich yet "low-quality" source domain (simulated data), and then fine-tuning the model towards the data-scarce yet "high-quality" target domain (experimental data). However, this vanilla workflow is challenged by two issues: a significant quality gap between source and target domains, and an extreme data scarcity of target data. Inspired by the recent success of sparsity regularizers, we propose to

incorporate sparsity to regularize both stages: sparsifying pre-training to improve the robustness and cross-domain transferability of learned features (Guo et al., 2018; Sehwag et al., 2019; Chen et al., 2022; Sehwag et al., 2020; Ding et al., 2022; Diffenderfer et al., 2021), and sparsifying fine-tuning to overcome data shortfalls (Liu et al., 2020; Chen et al., 2021; Tao et al., 2022). We demonstrate proof-of-concept experiments that even the simplest magnitude-based weight pruning could play effective regularization roles in our workflow. Furthermore, to avoid the ad-hoc two-step pruning as well as trial-and-error sparsity ratio selection at either stage, we propose a novel integrated optimization framework termed *Bi-Level Regularized Pre-training and Transfer* (**Bi-RPT**), that jointly learns optimal sparse masks and automatically allocates sparsity levels for both stages.

Our main contributions are summarized as follows:

- **Dataset:** We present **X-Yield**, the first public large-scale, multi-quality material science benchmark for HEA yield strength prediction, containing alloys' compositions, processing temperatures, and yield strengths. Specifically, the yield strengths of 240 HEAs are experimentally measured, while that of the remaining samples (over 100K) is calculated by simulations.

- **Methodology:** we formulate a cross-quality few-shot transfer workflow that can jointly exploit the simulated and experimental data for accurate predictions, and we innovate to leverage sparsity for addressing both the simulated/experimental domain gap and the scarcity of experimental data. While ad-hoc magnitude-based weight pruning is already found to be helpful, we further formulate an integrated bi-level optimization framework called **Bi-RPT** to automate the optimal sparse mask generation and sparsity ratio allocation at both pre-training and fine-tuning stages.

- **Results:** Extensive experiments show that Bi-RPT can boost performance on the *X-Yield* benchmark alongside other synthesized testbeds. In particular, for the yield strength regression task, we achieve a reduction of $19 \sim 38\%$ on the test mean squared error by merely using 5-10% of the available experimental data. For the yield strength classification task, we achieve $0.98\% \sim 1.53\%$ of improvement in terms of the test accuracy.

## 2 RELATED WORK

### 2.1 MACHINE LEARNING IN MATERIALS RESEARCH

ML has been applied to solve a wide range of problems in materials science ranging from the fields of inorganic chemistry (Kailkhura et al., 2019) to sustainability (Gomes et al., 2021), and metallurgy (Stan et al., 2020), with the typical purposes to predict materials properties and accelerate simulations (Pilania, 2021). In both cases, ML techniques are hailed as reducing computational time in contrast to traditional materials science methods and are typically fast to develop (Wei et al., 2019). Later on, deep learning has been successfully applied to problems in the field of HEAs, in particular to predict phase formation (Lee et al., 2021b; Zhu et al., 2022). These approaches provide significant increases in speed compared to phase predictions with CALculation of PHAse Diagrams (CALPHAD) (Saunders & Miodownik, 1998), density functional theory (Parr, 1983), and molecular dynamics methods (Shuichi, 1991) commonly used in materials science. Other properties predicted with deep learning are crystal structures, elastic constants (Liu et al., 2023) and hardness (Bhandari et al., 2021b). When it comes specifically to the yield strength of HEAs, its prediction has also been previously explored with deep learning (Liu et al., 2023; Bhandari et al., 2021a). However, a majority of these efforts are restricted to the development of specific alloys (Zheng et al., 2021; Bhandari et al., 2021b) or consist solely of transition metals (Wen et al., 2019), and many studies also only use a small experimental dataset for prediction (Wen et al., 2021). A generalized multi-fidelity ML model to predict yield HEA strength at scale remains to be absent yet highly demanded.

### 2.2 SPARSITY REGULARIZATION IN DEEP LEARNING

Sparsity or pruning was traditionally treated as a mainstream model compression approach in deep learning (Han et al., 2015). Recently, sparse regularizers have been increasingly used to enhance deep model robustness to various noise, malicious attacks, and distribution shifts. Guo et al. (2018); Sehwag et al. (2019); Gui et al. (2019) studied the intrinsic relationship between pruning and adversarial robustness. Recently, Diffenderfer et al. (2021) comprehensively demonstrated the benefit of model sparsification to improve robustness to distributional shifts (Hendrycks & Dietterich, 2019;

Bulusu et al., 2020). Sparse regularizers also exhibit promise in improving data efficiency. For example, Zheng et al. (2019); Liu et al. (2020) proposed to learn model pruning strategies for few-shot learning; Tian et al. (2020) combined model sparsification with meta-learning to improve few-shot performance. Sparse regularizers have even been proven effective beyond few-shot image classification, such as enhancing the data efficiency in image generation (Chen et al., 2021).

## 2.3 BI-LEVEL OPTIMIZATION

Bi-level optimization is a hierarchical framework where the variables in the *upper-level* optimization problem are dependent on the *lower-level* problem. Finn et al. (2017); Rajeswaran et al. (2019) formulated the meta-learning problem in the form of bi-level optimization, and solve it by using first-order approximations. Other applications of bi-level optimization include data and label poisoning (Mehra et al., 2021; Huang et al., 2020), and adversarial training (Zhang et al., 2021). In this work, we utilize bi-level optimization to formulate our two-stage workflow with sparsification and find each stage's optimal weights and sparse masks while considering their sequential dependency.

## 3 X-YIELD: A NEW BENCHMARK FOR HEA YIELD STRENGTH PREDICTION

**Overview**   Conventional alloys typically have one principal element with small amounts of other elements added to improve material properties (Ye et al., 2016) while HEAs can have multiple principal elements. The discovery of HEAs opened the door to a significantly wider range of design space to explore, most of which has yet to be examined (Miracle & Senkov, 2017). To address the task of using ML to predict HEA yield strength, we focus on the sub-field of refractory HEAs (RHEAs). These materials have been demonstrated to maintain excellent mechanical properties at high temperatures (Li et al., 2020), making them ideal candidates for hypersonics and aerospace industry applications. Prior work adopting ML to predict RHEA properties either uses solely experimental data (Wen et al., 2021), or restricts predictions to only transition metals (Wen et al., 2019) or specific alloys such as MoNbTaTiW (Bhandari et al., 2021a). Hence, a generalizable ML prediction model for a broad range of RHEAs is still absent. As mentioned earlier, it is impractical to generate high quantities of experimental data, especially for capturing yield strength at elevated temperatures. There are also challenges specific to high-temperature measurements such as controlling oxidation, confirming the heating profile and gradient within the samples, and use of more challenging experimental techniques (crosshead displacement) than those at lower temperatures (extensometers).

This work develops **X-Yield**, the first publicly available, multi-fidelity dataset consisting of over 100K low-quality simulated points and 240 experimental data points to explore the RHEA design space. In this study alone, the entire composition space of all alloys containing between ternary-septenary systems from the Al-Cr-Fe-Hf-Mo-Nb-Ta-Ti-V-W-Zr family is examined. Since obtaining real high-temperature yield strength data is challenging, a majority of the experimental yield strength data in the literature was taken close to room temperature (Borg et al., 2020) even though there is more interest in RHEA properties at the high-temperature end (Miracle & Senkov, 2017). From X-Yield, a multi-fidelity ML model is expected to be trained to predict high-temperature yield strength for a broad palette of RHEAs. The combination of high-temperature yield strengths from the simulated dataset and experimental input can generate an ML model to accurately and efficiently predict high-temperature yield strengths of alloys not included in the training set.

**Dataset Construction**   The yield strength of the simulation data was predicted using the analytic and parameter-free mechanistic yield strength model developed by Maresca & Curtin (2020). This model describes body-centered cubic (BCC) multi-principal element alloy (MPEA) solid solution strengthening associated with edge dislocations, in terms of elemental atomic volumes and elastic moduli. The yield strength was predicted for all ternary (1% increments), quaternary (1% increments), quinary (5% increments), senary (5% increments), and septenary (5% increments) alloys from the Al-Cr-Fe-Hf-Mo-Nb-Ta-Ti-V-W-Zr element family at temperatures between 300K-2500K in increments of 100K. This resulted in over three billion data points of which approximately 100,000 were randomly selected for inclusion in this study. Note that even this advanced simulation model suffers from notable oversimplification and data quality issues. For example, The phase stability and dislocation character were not used to filter alloys in the study and the model may overpredict the yield strength of alloys with non-BCC phases and underpredict the yield strength of alloys with different dislocation character, *e.g.* screw.

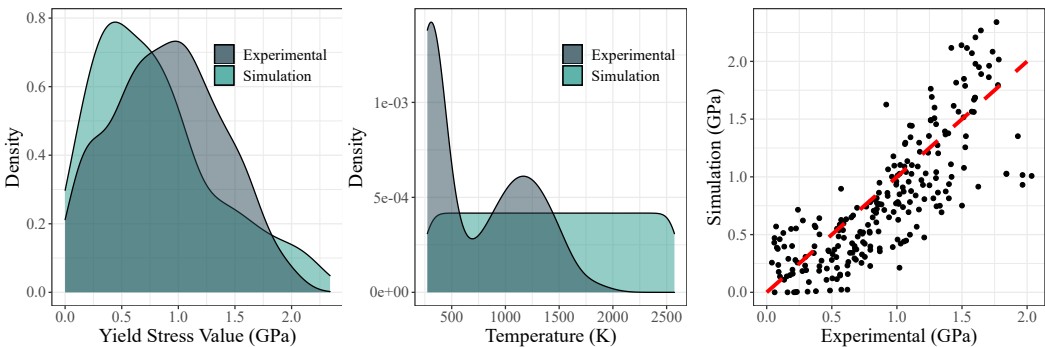

Figure 2: Left: the distribution of the yield stress; Middle: the distribution of the temperature; Right: pairwise visualization of the yield stress.

The high-quality experimental dataset was carefully filtered and curated from the database generated by Borg et al. (2020) consisting of mechanical property information for MPEAs. All data points were extracted that consisted solely of elements from the above element family, consisted only of BCC phases, were at temperatures higher than 20°C, and contained a yield strength value.

**Dataset Characteristics and "Quality Gap"**     As depicted in Figure 2, the simulation and experimental yield stress have different distributions. In the low-quality simulation data, a considerable portion of yield stress annotations is greater than 2, while the experimental data hardly contains yield stress points beyond 2 (with one datapoint exception) due to the experimental condition constraints. The distribution of the simulated yield stress is also significantly more skewed than the experimental ones. Pairwise visualization of the yield stress on the 240 high-quality experimental samples suggests a substantial deviation between the simulation and experimental results. The distributions of the processing temperatures are also heterogeneous, *i.e.*, the simulation data presents a uniform pattern while the temperatures in the conducted experiments are in a bimodal shape. These observations showcase the domain shifts or "quality gap" between simulations and experiments.

## 4    Cross-Quality Few-shot Transfer: A Two-Stage Workflow aided by Sparsity (Twice)

In this section, we first introduce the basic two-sage workflow, upon which we propose sparsification methods (a vanilla approach "Hand-Tune" and an improved principled framework "Bi-RPT").

**Basic Two-Stage Workflow: Pre-training then Fine-tuning**     Let us denote the high-quality target domain (experimental data) by $\mathcal{D}_t$, and the low-quality source domain (simulated data) by $\mathcal{D}_s$. Our goal is to learn a generalizable predictor over $\mathcal{D}_t$ while leveraging the aid of $\mathcal{D}_s$. One naive idea is to simply combine the two data domains and jointly train a supervised model. However, the large domain gap between $\mathcal{D}_s$ and $\mathcal{D}_t$, as well as the sample scarcity in $\mathcal{D}_t$, will result in the jointly trained predictor to fit $\mathcal{D}_t$ poorly. Instead, we propose to formulate our workflow as a two-stage pipeline: first pre-training a model on $\mathcal{D}_s$, and then fine-tuning to optimize the prediction over $\mathcal{D}_t$.

**Incorporating Bi-Stage Sparsity: A Vanilla Approach.**     However, the features learned from $\mathcal{D}_s$ will inevitably suffer from domain gap and noise when applied towards $\mathcal{D}_t$, and the extreme data scarcity of $\mathcal{D}_t$ remains as another challenge. Inspired by the recent success of sparse regularizers in improving both robustness/transferability and data efficiency, we attempt to incorporate sparsity into both stages to address the two-fold challenges.

We first prove our concepts by proposing a vanilla ad-hoc approach, which we refer to as *Hand-Tune*. Starting from pre-training over $\mathcal{D}_s$, we perform the standard iterative magnitude pruning (IMP) (Frankle & Carbin, 2019) during pre-training. In particular, we alternate between (re-)training and pruning; each time, we prune the 20% smallest-magnitude weights from the existing non-zero weights by default and continue (re-)training the remaining non-zero weights. Such a "prune-and-retrain" routine is repeated for $N_s$ rounds to obtain the final sparse mask $\boldsymbol{m}_s$ (1 denotes the element to be non-zero and 0 to be pruned) associated with the pretrained model weight. Then, we move on to fine-tuning over $\mathcal{D}_t$, and start another round of IMP on top of the pre-trained model: note that this second-stage IMP continues only on the subset of current non-zero weights, i.e., the 1-valued regions in $\boldsymbol{m}_s$. IMP in fine-tuning repeats another $N_t$ round (with the identical protocol as the first

stage), yielding another sparse mask $\boldsymbol{m}_t$. The final model uses the joint sparse mask $\boldsymbol{m}_s \odot \boldsymbol{m}_t$ where $\odot$ represents the point-wise product.

Hereby, $N_s$ and $N_t$ are hyperparameters that control the sparsity allocation between two stages. Intuitively, while certain sparsity may contribute to noise resilience, an overly large $N_s$ will cause the pre-trained model to be over-sparsified, limiting its capacity to learn sufficiently informative and transferable features. Fine-tuning has a similar trade-off. Therefore, $N_s$ and $N_t$ have to be manually tuned for the two-stage workflow to achieve good performance (see Appendix B.2).

**Principled Bi-Stage Sparsity Integration with Bi-RPT.** *Hand-Tune* has some apparent flaws: (1) it removes weight elements by merely using weight magnitude information, which is not explicitly task-driven; (2) the two sparse masks $\boldsymbol{m}_s$ and $\boldsymbol{m}_t$ are decided in a sequential manner rather than jointly optimized, e.g., learning $\boldsymbol{m}_t$ will passively suffer from any artifact in learning $\boldsymbol{m}_s$; (3) the sparsity ratios assigned in both stages, as controlled by $N_s$ and $N_t$, need to be manually tuned, without any obvious insight beyond exhaustive hyperparameter search.

We, therefore, devise a more principled framework that can jointly learn the optimal sparse masks as well as sparsity allocations for both stages, termed *Bi-Level Regularized Pre-training and Transfer* (**Bi-RPT**). The optimization problem is expressed as follows ($\gamma$ is a coefficient):

$$\min_{\boldsymbol{\theta}, \boldsymbol{m}_s, \boldsymbol{m}_t} \mathbb{E}_{(\boldsymbol{x}_t, y_t) \sim \mathcal{D}_t} \left[ \mathcal{L}_t((\boldsymbol{m}_s \odot \boldsymbol{m}_t) \odot \boldsymbol{\theta}, \boldsymbol{x}_t, y_t | \boldsymbol{\theta}^*, \boldsymbol{m}_s^*) \right] + \gamma \mathcal{R}(\boldsymbol{m}_s^* \odot \boldsymbol{m}_t) \tag{1}$$

$$\text{s.t. } \{\boldsymbol{\theta}^*, \boldsymbol{m}_s^*\} = \arg\min_{\boldsymbol{\theta}, \boldsymbol{m}_s} \mathbb{E}_{(\boldsymbol{x}_s, y_s) \sim \mathcal{D}_s} \mathcal{L}_s(\boldsymbol{m}_s \odot \boldsymbol{\theta}, \boldsymbol{x}_s, y_s), \tag{2}$$

where $\mathcal{L}_s / \mathcal{L}_t$ represents the objective functions for the two stages, respectively, $\boldsymbol{\theta}$ represents the models' parameters, and $\mathcal{R}$ represents the sparsity regularizer. Seemingly complicated at the first glance, the bi-level optimization formulation of Bi-RPT actually admits a clear physics "workflow" interpretation. Let us start from the *lower-level* problem (2) which instantiates the sparsity regularized pre-training stage over $\mathcal{D}_s$: its outputs include the pre-trained weight $\boldsymbol{\theta}^*$ and the corresponding sparse mask $\boldsymbol{m}_s$. Then, the *upper-level* problem (1) depicts the sparsity regularized fine-tuning over $\mathcal{D}_t$, which inherits both $\boldsymbol{\theta}^*$ and $\boldsymbol{m}_s^*$ as its starting point. It continues to modify the weight as well as to evolve another sparse mask $\boldsymbol{m}_t$. Eventually, a sparsity-promoting function $\mathcal{R}$ enforces the total sparsity over the joint mask $\boldsymbol{m}_s \odot \boldsymbol{m}_t$, and the final model weights could be represented as $(\boldsymbol{m}_s \odot \boldsymbol{m}_t) \odot \boldsymbol{\theta}$.

Importantly, the lower- and upper-level problems in Bi-RPT are solved in an end-to-end manner, meaning that even the fine-tuning depends on $\boldsymbol{\theta}^*$ and $\boldsymbol{m}_s^*$, it can, in turn, provide feedbacks for adjusting the latter: hence a synergistic optimization is achieved between two stages. The sparse mask selection now directly hinges on the end task (target domain loss $\mathcal{L}_t$) rather than heuristics such as weight magnitudes. Lastly, the sparsity levels of $\boldsymbol{m}_s$ and $\boldsymbol{m}_t$ do not need to be separately designated nor manually controlled: we automatically learn the sparsity ratio allocation, under only the total sparsity regularizer $\mathcal{R}$.

To practically solve the bi-level optimization of Bi-RPT, we derive algorithms whose details can be found in Appendix A. For the sparsity regularizer $\mathcal{R}$, we adopt the smoothed $\ell_0$ term (Guo et al., 2021) to facilitate differentiable training: a gate function $\boldsymbol{g}_\epsilon(\boldsymbol{x}) = \boldsymbol{x}^2/(\boldsymbol{x}^2 + \epsilon)$, whose outputs are almost binary when the $\epsilon$ is small, is used. In general, for the lower-level optimization problem, we update the models' parameters $\boldsymbol{\theta}$ by gradients to minimize $\mathcal{L}_s$; for the upper-level optimization, we utilize the gradient unrolling to develop update rules for $\boldsymbol{\theta}$.

### 4.1 PROOF-OF-CONCEPT EXPERIMENTS ON IMAGE DATA

For proof-of-concept, we conduct experiments on a synthesized testbed of image classification, to compare Hand-Tune and Bi-RPT. We adopt two source-domain dataset options: ImageNet (Deng et al., 2009) and ImageNet-C (Hendrycks & Dietterich, 2019), the latter more noisy and corrupted. Two target-domain options are also accompanied: CUB-200 (Wah et al., 2011) and *CUB-200 (10-shot)*, the latter designed to be rigorously "few-shot" where each class has only 10 training samples. Different combinations of $\mathcal{D}_s / \mathcal{D}_t$ allow us to conduct controlled experiments for stretch-testing various algorithm options' noise robustness as well as data efficiency.

Several baselines are compared to *Hand-Tune* and *Bi-RPT*: (1) *Pretrain-and-transfer*: the basic workflow of pre-training on $\mathcal{D}_s$ followed by finetuning on $\mathcal{D}_t$, with no sparsity involved; (2) *Pretrain*

*sparsity only /transfer sparsity only*: following our proposed pretrain-and-transfer workflow, but conducting IMP to only the pre-training/finetuning stage; (3) *No Pretraining*: directly training on $\mathcal{D}_t$ without using $\mathcal{D}_s$; (4) *Mix Training*: training one model on $\mathcal{D}_s$ and $\mathcal{D}_t$ combined. For those methods with IMP involved, we hand-select the sparsity ratio(s) for either or both stages that yield the highest generalization performance on $\mathcal{D}_t$, i.e., from hyperparameter grid search via cross-validation.

Table 1 reports the accuracies of all methods over various source/target combinations, on the same testing set of CUB-200 in Table 1. All methods use the same ResNet-18 backbone. We highlight several key observations: (1) incorporating $\mathcal{D}_s$ in general helps both CUB-200 and CUB-200 (10-shot), and the improvement margin is much more substantial for the few-shot case; (2) models trained by *Mix Training* fail to generalize on $\mathcal{D}_t$ - in fact even worse than *No Pretraining*, showcasing the negative influence of the quality gap; (3) in the same regime of pre-training then fine-tuning, adding appropriate sparsity helps, and two-stage sparsity can help more; (4) *Bi-RPT* stably outperforms *Hand-Tune* (especially, very notably in few-shot cases), despite the best efforts in tuning the latter's hyperparameters. More observations and analysis can be found in Appendix B (Tables A5 - A8, and Figure A4): including but unlimited to the backfiring effect of "over-sparsification", and the compound influence of per-stage IMP sparsity allocation in *Hand-Tune*.

Table 1: Experiments on image data: testing accuracy of fine-tuned ResNet-18 on CUB-200 / CUB-200 (10-shot) as $\mathcal{D}_t$, after pretraining on ImageNet and ImageNet-C as $\mathcal{D}_s$, respectively.

| $\mathcal{D}_t$ | Methods | Two-stage | $m_s$ | $m_t$ | $\mathcal{D}_s$ | |
|---|---|---|---|---|---|---|
| | | | | | ImageNet | ImageNet-C |
| CUB-200 / CUB-200 (10-shot) | No Pretraining | ✗ | ✗ | ✗ | 44.27% / 7.98% | |
| | Mix Training | ✗ | ✗ | ✗ | 30.88% / 6.72% | 27.32%/6.89% |
| | Pretrain-and-transfer | ✓ | ✗ | ✗ | 74.16% / 38.66% | 71.59% / 32.14% |
| | Pretrain sparsity only | ✓ | ✓ | ✗ | 76.01% / 40.73% | 73.70% / 38.76% |
| | Transfer sparsity only | ✓ | ✗ | ✓ | 74.16% / 38.90% | 71.83% / 32.53% |
| | Hand-Tune | ✓ | ✓ | ✓ | 76.01% / 40.78% | 74.01% / 39.94% |
| | Bi-RPT | ✓ | ✓ | ✓ | **78.60% / 51.55%** | **76.29% / 47.01%** |

## 5 MAIN EXPERIMENTS ON THE X-YIELD BENCHMARK

### 5.1 IMPLEMENTATION DETAILS

**Task Definition.** The most naturally defined task on X-Yield is the regression, *i.e.*, predicting the yield strength of alloys, and calculating the error between the model prediction and "ground-truth" (experimental results). Besides the regression task, we formulate another surrogate classification task by constructing five categorical labels based on the bin intervals where the ground-truth yields strength fall in. These intervals are: $[0, 0.5), [0.5, 1), [1, 1.5), [1.5, 2),$ and $[2, \infty)$.

**Data Representations.** We featurize each HEA by mapping its composition and temperature into a "pseudoimage" (please refer to Appendix B.5 and Figure A5). The pseudoimages have two channels: the first channel is constructed from the alloys' composition using the randomized periodic table structure (Feng et al., 2021). As the temperatures are originally recorded in Kelvin, we convert and normalize them by $T_{\text{normalized}} = (K - 273.15)/2000$ where $K$ is the temperature in Kelvin, and then embed the converted temperature as the second channel in pseudoimages.

**Architectures and Baselines.** The structure of the ML predictor we use is a convolutional neural network. It consists of 3 convolutional layers, each of which has a kernel size of 3, followed by Batch Normalization (Ioffe & Szegedy, 2015) and ReLU (Glorot et al., 2011) activation. A multilayer perceptron is appended after the convolutional neural network to generate the final prediction for both the regression and classification tasks. We focus on comparing our main proposal, Bi-RPT, with two baselines of *No Pretraining* and *Pretrain-and-transfer*, same as defined in Section 4.1.

**Evaluation Metrics and Data Splits.** We evaluate each method in two ways. Besides the widely used 10-fold cross-validation, we explore two challenging *extreme few-shot* settings: we sample 5% (and 10%) of experimental data from each alloy type (ternary, quaternary, quinary and senary) as our training set, and the rest are left as the test set. Note that these classes *are not* the classification

labels. Eventually, we have only 23(11) training samples and 217(229) testing samples. All the low-quality (simulated) data is used for pretraining where applicable. For the regression task, we report the best mean squared error (MSE) on the test splits; and for the classification task, we report models' test split accuracy to measure their performance.

**Training Hyperparameters** We pretrain the ML predictor on the simulation data for 10 epochs. During the pretraining, we use the Adam optimizer (Kingma & Ba, 2014) with an initial learning rate of $1 \times 10^{-4}$ and a cosine annealing schedule (Loshchilov & Hutter, 2016). For the transfer stage, we fine-tune the pretrained model on the experimental data for 90 epochs. The optimizer we use is the SGD optimizer with an initial learning rate of $1 \times 10^{-3}$. We also decay the learning rate by 10 for every 30 epoch. The batch sizes for pretraining and fine-tuning are 16 and 4, respectively.

## 5.2 MAIN RESULTS

**Classification and regression with extreme few-shot settings.** We first apply Bi-RPT to solve the regression and classification tasks under the two extreme few-shot settings where only 5% and 10% experimental data are available, respectively. Table 2 shows that: (1) pretraining on simulation data can benefit the ML predictor consistently on both the regression (over 10% reduction in MSE) and classification (over 11% improvement in accuracy) tasks, especially when the data is more scarce; (2) the integration of sparsity into the pretraining and transfer workflow can further strengthen the predictor's generalization, improving accuracy by 0.98% and reducing MSE by 8.91% using merely 10% training experiment data, and the improvement also becomes even more significant with 5% training data (1.53% increase in terms of the accuracy and 19.75% reduction in terms of the MSE).

Table 2: Test accuracy on the testing set of different splits of high-fidelity alloy data. The experiments are repeated 10 times, and we report both the mean and the 95% confidence interval.

| Method | 10% train samples | | 5% train samples | |
|---|---|---|---|---|
| | Test MSE | Test Accuracy | Test MSE | Test Accuracy |
| No Pretraining | $0.114 \pm 0.007$ | $54.84 \pm 1.59\%$ | $0.212 \pm 0.041$ | $47.25 \pm 0.80\%$ |
| Pretrain-and-transfer | $0.101 \pm 0.001$ | $65.85 \pm 0.88\%$ | $0.162 \pm 0.002$ | $62.84 \pm 1.96\%$ |
| Bi-RPT | $\mathbf{0.092 \pm 0.011}$ | $\mathbf{66.83 \pm 1.41\%}$ | $\mathbf{0.130 \pm 0.006}$ | $\mathbf{64.37 \pm 1.10\%}$ |

**Classification and regression with 10-fold cross-validation.** On the slightly "data-rich" 10-fold cross-validation setting, we have observed a similar trend: the bi-stage regime of pretraining and transfer out-performs the single-stage training pipeline, and incorporating sparsity can consistently provide remarkable improvement to the ML predictor, particularly in the regression performance.

Table 3: Classification and regression performance under the ten-folded cross-validation settings.

| | Classification | Regression |
|---|---|---|
| No Pretraining | $67.50 \pm 5.16\%$ | $0.226 \pm 0.027$ |
| Pretrain-and-transfer | $82.09 \pm 3.86\%$ | $0.206 \pm 0.026$ |
| Bi-RPT | $\mathbf{82.50 \pm 2.93\%}$ | $\mathbf{0.068 \pm 0.009}$ |

**Performance comparison on alloys at various temperatures.** Based on the trained model with 10% experimental data, we predict the yield strength of three alloys, MoNbTaTi, MoNbTaTiW and HfMoNbTaTiZr, at different temperatures. Table 4 shows the predicted yield stress on these three alloys using Bi-RPT and baselines. On the quinary and senary alloy systems, Bi-RPT shows exceptional precision in predicting the experimental yield stress. More scrutiny of those predictions reveals several findings that neatly align with our material science expertise. For example, it is known that screw dislocations are more likely to be dominant than the edge in MoNbTi and NbTaTi ternaries (shown from the ternary comparison in the Citrine database (Borg et al., 2020)). Thus it makes sense that the model under-predicts the MoNbTaTi and MoNbTaTiW cases: our model seems to correctly pick up these differences and predicts a higher yield strength. Another example is that our model over-predicts HfMoNbTaTiZr at lower temperatures (300K ∼ 900K). Since all our collected experimental samples are 100% body-centered cubic (which shows, admittedly, a limitation

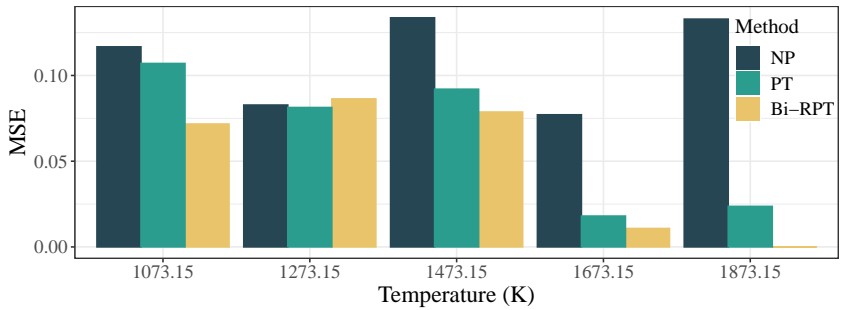

Figure 3: Prediction MSE under different temperatures. We compare the results of three methods: No Pretraining (NP), Pretrain-and-transfer (PT), and Bi-RPT.

of X-Yield compared to the tremendous variations in real-world HEAs), it is likely that a non-BCC phase will appear at lower temperatures, hence lowering the yield strength.

Table 4: Predicted yield stress of different alloys under different temperatures. Only 10% of the experimental data are available during fine-tuning. We compare the predicted yield stress generated by Bi-RPT with our "No Pretraining" (NP) and "Pretrain-and-transfer" (PT) baselines and the simulation. The numbers with the smallest error are marked in **bold**.

| Alloys | Temperature (K) | Predicted Yield Stress (GPa) | | | | Experimental (GPa) |
|---|---|---|---|---|---|---|
| | | Bi-RPT | NP | PT | Simulation | |
| MoNbTaTi | 293.15 | 1.078 | **1.170** | 1.062 | 0.475 | 1.210 |
| | 473.15 | **0.965** | 1.004 | 0.902 | 0.381 | 0.868 |
| | 673.15 | 0.746 | 0.772 | **0.731** | 0.282 | 0.685 |
| | 873.15 | 0.508 | 0.642 | **0.584** | 0.472 | 0.593 |
| | 1273.15 | 0.425 | **0.570** | 0.488 | 0.114 | 0.539 |
| MoNbTaTiW | 298.15 | **1.268** | 1.031 | 1.068 | 0.814 | 1.399 |
| | 873.15 | **0.677** | 0.569 | 0.607 | 0.372 | 0.689 |
| | 1073.15 | **0.618** | 0.520 | 0.523 | 0.294 | 0.674 |
| | 1273.15 | **0.536** | 0.530 | 0.486 | 0.232 | 0.620 |
| HfMoNbTaTiZr | 296.15 | **1.527** | 1.051 | 1.132 | 1.849 | 1.515 |
| | 873.15 | **0.861** | 0.556 | 0.685 | 1.178 | 0.973 |
| | 1073.15 | **0.762** | 0.536 | 0.612 | 0.516 | 0.791 |
| | 1273.15 | **0.662** | 0.563 | 0.573 | 0.421 | 0.753 |

**Performance at high temperatures.** One of the important tasks in the alloy design community is to find alloys that are capable of withstanding stress at high temperatures. To verify if Bi-RPT can provide reliable recommendations to help the community achieve this goal, we look deeper into the predictive performance in high-temperature regimes. We train our model with 10% data, predict the yield stress for the rest 90%, and compare the predictive quality of models at high temperatures in Figure 3. We can see that Bi-RPT significantly outperforms other baselines, especially at temperatures greater than 1400K. These results suggest Bi-RPT could serve as a strong tool for designing HEAs with superior yield stress at elevated temperatures.

## 6 CONCLUSIONS

To address the important yet challenging problem of HEA yield stress prediction, we curated and released X-Yield, the first large-scale, multi-fidelity benchmark. To effectively leverage this benchmark, we also designed a two-stage cross-quality few-shot transfer workflow and proposed to utilize sparsity to tackle both challenges of low data quality at pretraining and scarcity at transfer. Besides ad-hoc methods, we formulated a principled bi-level optimization framework to automatically learn the optimal sparse masks and sparsity allocation between two stages. Extensive experiments on both image data testbeds and X-Yield demonstrate the Bi-RPT showed a substantial improvement over existing baselines. Moving forward, we are now closely working with material scientists to validate our ML prediction results based on their domain expertise, and the team has already identified some alloy candidates that appear promising to be experimentally validated.

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

# A    MORE DETAILS ON METHODS

In this section, we present the technical details of our proposed method and framework ("Hand-Tune" and "Bi-RPT").

## A.1    HAND-TUNE

Hand-Tune decides sparse masks for the two stages in an iterative way as explained in Algorithm 1.

---

**Algorithm 1** Hand-Tune

---

**Input:** Initialization weights $\boldsymbol{\theta}_0$, low-quality pretraining dataset $\mathcal{D}_s$, high-quality fine-tuning dataset $\mathcal{D}_t$, number of IMP rounds $N_s$ for the pretraining stage and $N_t$ for the fine-tuning stage.
**Output:** the trained weights $\boldsymbol{\theta}^*$, the sparse mask $\boldsymbol{m}_s$ for the pretraining stage, and the sparse mask $\boldsymbol{m}_t$ for the fine-tuning stage.
Initialize the sparse masks $\boldsymbol{m}_s$ for the pretraining stage to be a all "1" mask.
Initialize the model's weight as $\boldsymbol{\theta}_0$ and train the weights on $\mathcal{D}_s$ to obtain $\boldsymbol{\theta}_s$.
**for** $i = 1, 2, \ldots, N_s$ **do**                                   ▷ IMP at the pre-training stage
    Prune 20% of the smallest-magnitude weights from the non-zero regions of $\boldsymbol{m}_s \odot \boldsymbol{\theta}_s$, by setting the values at corresponding positions to those weights in $\boldsymbol{m}_s$ to "0".
    (Re-)train the sparse weights $\boldsymbol{m}_s \odot \boldsymbol{\theta}_s$ on $\mathcal{D}_s$. Only $\boldsymbol{\theta}_s$ is updated.
**end for**
Initialize the sparse masks at the fine-tuning stage $\boldsymbol{m}_t$ to be all "1" masks and freeze $\boldsymbol{m}_s$.
Initialize model's weight as $\boldsymbol{m}_s \odot \boldsymbol{\theta}_s$, and train on $\mathcal{D}_t$ to obtain $\boldsymbol{m}_s \odot \boldsymbol{\theta}_t$.
**for** $i = 1, 2, \ldots, N_t$ **do**                                   ▷ IMP at the fine-tuning stage
    Prune 20% of the smallest-magnitude weights from the non-zero regions of weights $(\boldsymbol{m}_s \odot \boldsymbol{m}_t) \odot \boldsymbol{\theta}_s$, by setting the values at corresponding positions to those weights in $\boldsymbol{m}_t$ to "0".
    (Re-)train the sparse weights $(\boldsymbol{m}_s \odot \boldsymbol{m}_t) \odot \boldsymbol{\theta}_t$ on $\mathcal{D}_t$. Only $\boldsymbol{\theta}_t$ is updated.
**end for**
Obtain the final sparse weights $(\boldsymbol{m}_s \odot \boldsymbol{m}_t) \odot \boldsymbol{\theta}^*$ and return $\boldsymbol{\theta}^*$, $\boldsymbol{m}_s$ and $\boldsymbol{m}_t$.

---

## A.2    BI-RPT

We now build the techniques to solve the bi-level optimization problem formulated in Bi-RPT.

**Formulation**

$$\min_{\boldsymbol{\theta}, \boldsymbol{m}_s, \boldsymbol{m}_t} \mathbb{E}_{(\boldsymbol{x}_t, y_t) \sim \mathcal{D}_t} \left[ \mathcal{L}_t((\boldsymbol{m}_s \odot \boldsymbol{m}_t) \odot \boldsymbol{\theta}, \boldsymbol{x}_t, y_t | \boldsymbol{\theta}^*, \boldsymbol{m}_s^*) \right] + \gamma \mathcal{R}(\boldsymbol{m}_s^* \odot \boldsymbol{m}_t)$$

$$\text{s.t. } \{\boldsymbol{\theta}^*, \boldsymbol{m}_s^*\} = \arg\min_{\boldsymbol{\theta}, \boldsymbol{m}_s} \mathbb{E}_{(\boldsymbol{x}_s, y_s) \sim \mathcal{D}_s} \mathcal{L}_s(\boldsymbol{m}_s \odot \boldsymbol{\theta}, \boldsymbol{x}_s, y_s).$$

**Lower-level problem**    We solve the lower-level problem through a $p$-step SGD unrolling. Let $\boldsymbol{\theta}^{(k)}$ be the model weights, and $\boldsymbol{m}_s^{(k)}$ be the mask for the pretraining stage. The superscript $(k)$ indicates they have been updated on the upper-level for $k$ steps.

$\boldsymbol{\theta}^{(k)}$ and $\boldsymbol{m}_s^{(k)}$ will be the starting points for the lower-level optimization problem. $\boldsymbol{\theta}_l^{(t)}$ and $\boldsymbol{m}_{s,l}^{(t)}$ are the weights and mask, respectively, after being updated for $t$ steps on the lower-level optimization problem (implying $\boldsymbol{\theta}_l^{(0)} = \boldsymbol{\theta}^{(k)}$ and $\boldsymbol{m}_{s,l}^{(0)} = \boldsymbol{m}_s^{(k)}$). The update rules can be written as

$$\boldsymbol{\theta}_l^{(0)} = \boldsymbol{\theta}^{(k)}, \boldsymbol{\theta}_l^{(p)} = \boldsymbol{\theta}_l^{(p-1)} - \lambda_l \nabla_{\boldsymbol{\theta}} \mathcal{L}_s |_{\boldsymbol{\theta} = \boldsymbol{\theta}_l^{(p-1)}}, \tag{3}$$

$$\boldsymbol{m}_{s,l}^{(0)} = \boldsymbol{m}_s^{(k)}, \boldsymbol{m}_{s,l}^{(p)} = \boldsymbol{m}_{s,l}^{(p-1)} - \lambda_{m,l} \nabla_{\boldsymbol{m}} \mathcal{L}_s |_{\boldsymbol{m} = \boldsymbol{m}_{s,l}^{(p-1)}}, \tag{4}$$

where $\lambda_l$ is the learning rate for the model weight $\boldsymbol{\theta}$, and $\lambda_{m,l}$ is the learning rate for the mask $\boldsymbol{m}_{s,l}^{(t)}$ at the lower-level optimization problem.

**Upper-level problem and Sparse Regularization Loss**   The upper-level problem is the sum of two losses: a normal training loss $\mathcal{L}_t$ and a sparse regularization loss $\mathcal{R}$ ($\gamma$ is a coefficient).

We first develop update rules for the training loss $\mathcal{L}_t$. The weights $\boldsymbol{\theta}^*(:= \boldsymbol{\theta}_l^{(p)})$ and masks $\boldsymbol{m}_s^*(:= \boldsymbol{m}_{s,l}^{(p)})$ from the lower-level problem after $p$ unroll steps will serve as the initialization of the upper-level problem. We update the model weight $\boldsymbol{\theta}$ and masks at the upper level by applying gradient-based methods (take SGD as an example):

$$
\begin{aligned}
\boldsymbol{\theta}^{(k+1)} &= \boldsymbol{\theta}^* - \lambda_u \frac{\mathrm{d}\mathcal{L}_t}{\mathrm{d}\boldsymbol{\theta}^*} \\
&= \boldsymbol{\theta}^* - \lambda_u \Big( \frac{\partial \mathcal{L}_t}{\partial \boldsymbol{\theta}^*} + \frac{\partial \mathcal{L}_t}{\partial \boldsymbol{m}_s^*} \frac{\partial \boldsymbol{m}_s^*}{\partial \boldsymbol{\theta}^*} \Big),
\end{aligned}
\tag{5}
$$

where $\lambda_u$ is the learning rate for the weights for the upper-level optimization problem. The gradient on $\boldsymbol{m}_t$ is easy enough: $\frac{\partial \mathcal{L}_t}{\partial \boldsymbol{m}_t}$, while the gradient on $\boldsymbol{m}_s$ is slightly complicated:

$$
\frac{\mathrm{d}\mathcal{L}_t}{\mathrm{d}\boldsymbol{m}_s^*} = \frac{\partial \mathcal{L}_t}{\partial \boldsymbol{m}_s^*} + \frac{\partial \mathcal{L}_t}{\partial \boldsymbol{\theta}^*} \frac{\partial \boldsymbol{\theta}^*}{\partial \boldsymbol{m}_s^*}.
\tag{6}
$$

We expand the latter terms in Eqn. 5 and Eqn. 6 based on the first-order approximation (picking $p = 1$) on the lower-level problem:

$$
\begin{aligned}
\frac{\partial \boldsymbol{\theta}^*}{\partial \boldsymbol{m}_s^*} = \frac{\partial(\boldsymbol{\theta}_l^{(0)} - \lambda_l \nabla_{\boldsymbol{\theta}}\mathcal{L}_s)}{\partial(\boldsymbol{m}_{s,l}^{(0)} - \lambda_{m,l}\nabla_{\boldsymbol{m}_s}\mathcal{L}_s)} &= \frac{\partial(\boldsymbol{\theta}_l^{(0)} - \lambda_l \nabla_{\boldsymbol{\theta}}\mathcal{L}_s)}{\partial\boldsymbol{\theta}_l^{(0)}} \frac{\partial\boldsymbol{\theta}_l^{(0)}}{\partial(\boldsymbol{m}_{s,l}^{(0)} - \lambda_{m,l}\nabla_{\boldsymbol{m}_s}\mathcal{L}_s)} + \\
& \quad \frac{\partial(\boldsymbol{\theta}_l^{(0)} - \lambda_l \nabla_{\boldsymbol{\theta}}\mathcal{L}_s)}{\partial\boldsymbol{m}_{s,l}^{(0)}} \frac{\partial\boldsymbol{m}_{s,l}^{(0)}}{\partial(\boldsymbol{m}_{s,l}^{(0)} - \lambda_{m,l}\nabla_{\boldsymbol{m}_s}\mathcal{L}_s)} \\
&= (\mathrm{I} - \lambda_l\nabla_{\boldsymbol{\theta}}^2\mathcal{L}_s)(-\lambda_{m,l}\nabla_{\boldsymbol{m}_s\boldsymbol{\theta}}\mathcal{L}_s)^{-1} + \\
& \quad (-\lambda_l\nabla_{\boldsymbol{m}_s\boldsymbol{\theta}}\mathcal{L}_s)(\mathrm{I} - \lambda_{m,l}\nabla_{\boldsymbol{m}_s}^2\mathcal{L}_s)^{-1}, \\
\frac{\partial\boldsymbol{m}_s^*}{\partial\boldsymbol{\theta}^*} = \frac{\partial(\boldsymbol{m}_{s,l}^{(0)} - \lambda_{m,l}\nabla_{\boldsymbol{m}_s}\mathcal{L}_s)}{\partial(\boldsymbol{\theta}_l^{(0)} - \lambda_l\nabla_{\boldsymbol{\theta}}\mathcal{L}_s)} &= (\mathrm{I} - \lambda_l\nabla_{\boldsymbol{\theta}}^2\mathcal{L}_s)^{-1}(-\lambda_{m,l}\nabla_{\boldsymbol{m}_s\boldsymbol{\theta}}\mathcal{L}_s) + \\
& \quad (-\lambda_l\nabla_{\boldsymbol{m}_s\boldsymbol{\theta}}\mathcal{L}_s)^{-1}(\mathrm{I} - \lambda_{m,l}\nabla_{\boldsymbol{m}_s}^2\mathcal{L}_s).
\end{aligned}
$$

$$\tag{7}$$
$$\tag{8}$$

Further approximations can be made to avoid the matrix inverse and save computation:

$$
\frac{\partial\boldsymbol{\theta}^*}{\partial\boldsymbol{m}_s^*} \approx -\lambda_l\nabla_{\boldsymbol{m}_s\boldsymbol{\theta}}\mathcal{L}_s \quad , \frac{\partial\boldsymbol{m}_s^*}{\partial\boldsymbol{\theta}^*} \approx -\lambda_{m,l}\nabla_{\boldsymbol{m}_s\boldsymbol{\theta}}\mathcal{L}_s.
$$

Based on the rules, $\boldsymbol{m}_s$ and $\boldsymbol{m}_t$ can be optimized by:

$$
\hat{\boldsymbol{m}}_t^{(k+1)} = \boldsymbol{m}_t^{(k)} - \lambda_m \frac{\partial\mathcal{L}_t}{\partial\boldsymbol{m}_t}\Big|_{\boldsymbol{m}_t=\boldsymbol{m}_t^{(k)}}, \quad \hat{\boldsymbol{m}}_s^{(k+1)} = \boldsymbol{m}_s^{(k)} - \lambda_m \frac{\partial\mathcal{L}_t}{\partial\boldsymbol{m}_s} + \lambda_m\lambda_l \frac{\partial\mathcal{L}_t}{\partial\boldsymbol{\theta}^*}\nabla_{\boldsymbol{m}_s\boldsymbol{\theta}}\mathcal{L}_s\Big|_{\boldsymbol{m}_s=\boldsymbol{m}_s^{(k)}},
\tag{9}
$$

where the superscript $(k)$ means the steps updated.

We then focus on the latter term. We choose $\ell_0$ loss (*i.e.* the number of non-zero elements) as the sparse regularizer $\mathcal{R}$, which is not differentiable and difficult to optimize. Therefore, we follow Guo et al. (2021) to use the smoothed $\ell_0$ formulation to facilitate differentiable training. Specifically, a gate function $g_\epsilon(\boldsymbol{x}) := \frac{\boldsymbol{x}^2}{\boldsymbol{x}^2+\epsilon}$, where $\epsilon$ is a small positive number, is used to replace the binary masks, which are instead parameterized by $g_\epsilon(\boldsymbol{m}_s)$ and $g_\epsilon(\boldsymbol{m}_t)$. We decay the value of $\epsilon$ every

epoch, and the gate function will gradually output only polarized numbers (*i.e.,* 0 and 1). We further apply the proximal-SGD (Nitanda, 2014) to minimize the $\ell_0$ loss: after we update the $\boldsymbol{m}_s$ and $\boldsymbol{m}_t$ with respect to $\mathcal{L}_t$ by gradient descent-based methods (Eqn. 9), we use the proximal operator to alternatively update each mask. For $\boldsymbol{m}_s$, the formulation can be written as:

$$\text{prox}_{\lambda_m \gamma \mathcal{R}}(\boldsymbol{m}_s^{(k+1)}) = \arg\min_{\boldsymbol{m}_s} \frac{1}{2} \|\boldsymbol{m}_s^{(k+1)} \odot \hat{\boldsymbol{m}}_t^{(k+1)} - \hat{\boldsymbol{m}}_s^{(k+1)} \odot \hat{\boldsymbol{m}}_t^{(k+1)}\|_2^2 + \lambda_m \gamma \|\boldsymbol{m}_s^{(k+1)} \odot \hat{\boldsymbol{m}}_t^{(k+1)}\|_0.$$

We follow (Guo et al., 2021) to solve it by relaxing it to the $\ell_1$ norm problem, which has a closed form solution:

$$m_{s,i} = \begin{cases} \hat{m}_{s,i}^{(k+1)} - \frac{\gamma \lambda_m}{\hat{m}_{t,i}^{(k+1)}}, & \hat{m}_{s,i}^{(k+1)} \geq \frac{\gamma \lambda_m}{\hat{m}_{t,i}^{(k+1)}} \\ \hat{m}_{s,i}^{(k+1)} + \frac{\gamma \lambda_m}{\hat{m}_{t,i}^{(k+1)}}, & \hat{m}_{s,i}^{(k+1)} \leq -\frac{\gamma \lambda_m}{\hat{m}_{t,i}^{(k+1)}} \\ 0, & -\frac{\gamma \lambda_m}{\hat{m}_{t,i}^{(k+1)}} < \hat{m}_{s,i}^{(k+1)} < \frac{\gamma \lambda_m}{\hat{m}_{t,i}^{(k+1)}} \end{cases}, \tag{10}$$

where $m_{s,i}$ is the $i$-th element in $\boldsymbol{m}_s$ (the same for $m_{t,i}$).

Similarly, we derive the update for $\boldsymbol{m}_t$:

$$m_{t,i}^{(k+1)} = \begin{cases} \hat{m}_{t,i}^{(k+1)} - \frac{\gamma \lambda_m}{\hat{m}_{s,i}^{(k+1)}}, & \hat{m}_{t,i}^{(k+1)} \geq \frac{\gamma \lambda_m}{\hat{m}_{t,i}^{(k+1)}} \\ \hat{m}_{t,i}^{(k+1)} + \frac{\gamma \lambda_m}{\hat{m}_{s,i}^{(k+1)}}, & \hat{m}_{t,i}^{(k+1)} \leq -\frac{\gamma \lambda_m}{\hat{m}_{t,i}^{(k+1)}} \\ 0, & -\frac{\gamma \lambda_m}{\hat{m}_{s,i}^{(k+1)}} < \hat{m}_{t,i}^{(k+1)} < \frac{\gamma \lambda_m}{\hat{m}_{s,i}^{(k+1)}} \end{cases}. \tag{11}$$

Finally, we combine all these components into Algorithm 2.

---

**Algorithm 2** Solving Bi-RPT

---

**Input:** Initialization weights $\boldsymbol{\theta}_0$, training loss functions for two stages $\mathcal{L}_s$ and $\mathcal{L}_t$, low-quality pretraining dataset $\mathcal{D}_s$, high-quality fine-tuning dataset $\mathcal{D}_t$, number of steps for gradient unroll $p$.
**Output:** Trained model weights $\boldsymbol{\theta}$, sparse masks $\boldsymbol{m}_s$ and $\boldsymbol{m}_t$.
Train $\boldsymbol{\theta}_0$ on $\mathcal{D}_s$ to get weights $\boldsymbol{\theta}$.
**while** not converged **do**
    Given the fixed $\boldsymbol{m}_s$, update the weights $\boldsymbol{\theta}$ on $\mathcal{D}_s$ by gradient unrolling (Eqn. 3)
    Update the weights $\boldsymbol{\theta}$ by Eqn. 5
    Update the masks $\boldsymbol{m}_s$ and $\boldsymbol{m}_t$ by Eqn. 9.
    Update the masks $\boldsymbol{m}_s$ and $\boldsymbol{m}_t$ by Eqn. 10 and Eqn. 11.
**end while**

---

# B MORE EXPERIMENTS DETAILS AND RESULTS

## B.1 BASELINES AND HYPERPARAMETERS

We list the hyper-parameters we used for all the baselines in this section.

**General Settings.** When pre-training the models on $\mathcal{D}_s$ (ImageNet and ImageNet-C), we use the SGD optimizer and a learning rate is $4 \times 10^{-1}$. We linearly warm-up the learning rate within 5 epochs, and then decay it by 10 for every 30 epochs. Models are pretrained for 95 epochs on $\mathcal{D}_s$, with a batch size of 1024. On $\mathcal{D}_t$, *i.e.*, CUB-200 and CUB-200 (10-shot), we set the initial learning rate as $1 \times 10^{-3}$. The learning rate is decayed by 10 every 30 epochs, and the model is trained for 90 epochs with a batch size of 64.

For Hand-Tune, we train the models with 95 epochs from scratch on $\mathcal{D}_s$ to get a *densely* pretrained models. The number of training epochs is reduced to 45 after the pretrained model is derived. After the pretraining stage ends, we continue to transfer the model on $\mathcal{D}_t$ following the above hyperparameters. The number of epochs is also reduced to 45 after we prune the weights.

For No-Pretraining, we train the model using an initial learning rate of $1 \times 10^{-2}$ and a batch size of 64. For Mix-Training, as the number of classes is different for ImageNet and CUB-200, we use

two fully-connected layers on top the normal ResNet-18 backbone, and train them simultaneously. We sample batches from the two domains ($\mathcal{D}_s$ and $\mathcal{D}_t$) using the same batch size of 64. The initial learning rate for these methods are $1 \times 10^{-2}$, and it is decayed by 10 every 30 epochs.

For Bi-RPT, we follow the same learning rate settings despite some additional hyper-parameters are newly introduced. The learning rate for the lower-level problem ($\lambda_l$) is $1 \times 10^{-3}$, the same as the learning rate for upper-level problem ($\lambda_u$). The value of $\gamma$ is set to $1 \times 10^{-4}$, which are determined through ablation studies in Table A10. The value of $\lambda_m$ are set to 3.5, which are also determined through ablation studies in Table A11.

## B.2 PERFORMANCE OF HAND-TUNE UNDER DIFFERENT LEVELS OF SPARSITY

We report the performance of the Hand-Tune method under different levels of sparsity. We conduct experiments with $N_s = \{0, 1, 2, 3, 4, 5\}$ and $N_t = \{0, 1, 2, 3, 4\}$, resulting sparsity levels at pre-training stage of $\{0.00\%, 20.00\%, 36.00\%, 48.80\%, 59.04\%, 67.23\%\}$ and sparsity levels at transfer stage of $\{0.00\%, 20.00\%, 36.00\%, 48.80\%, 59.04\%\}$. We conduct experiments over all the combinations of pretraining and transfer pruning rounds. More specifically, we first perform IMP on $\mathcal{D}_s$ for $N_s$ rounds, and continue to perform IMP on $\mathcal{D}_t$ for another $N_t$ rounds. The experiment results over various source and target combinations are shown in Table A5 to Table A8. Note that all the models are evaluated on the testing samples in $\mathcal{D}_t$.

From this series of tables we observe that: (1) sparsity at pretraining helps improve the model's performance on $\mathcal{D}_t$ after fine-tuning, and the performance gain is larger when $\mathcal{D}_s$ contains more noise and has larger domain shifts; (2) sparsity at transfer is also beneficial to the performance after fine-tuning, and the improvement is more significant when the $\mathcal{D}_t$ is more "data-scarce"; (3) the optimal sparse levels for the two stages vary for different combinations of pretrain and transfer domains, highlighting the importance of choosing the correct pruning rounds for both stages.

Table A5: Test accuracy of fine-tuned ResNet-18 on CUB-200 after pretrained on ImageNet, under different levels of sparsity at pretraining and sparsity at transfer.

| Sparsity At Transfer | Sparsity At Pretraining | | | | | |
| --- | --- | --- | --- | --- | --- | --- |
| | 0.00% | 20.00% | 36.00% | 48.80% | 59.04% | 67.23% |
| 0.00% | 74.16% | **76.01%** | 75.77% | 75.87% | 74.99% | 74.35% |
| 20.00% | 74.15% | 75.54% | 75.82% | 75.98% | 74.73% | 74.46% |
| 36.00% | 74.13% | 75.08% | 75.56% | 75.73% | 74.06% | 73.94% |
| 48.80% | 73.84% | 74.01% | 74.56% | 74.46% | 72.37% | 72.16% |
| 59.04% | 73.61% | 73.77% | 73.61% | 72.89% | 70.66% | 70.56% |

Table A6: Test accuracy of fine-tuned ResNet-18 on CUB-200 after pretrained on ImageNet-C, under different levels of sparsity at pretraining and sparsity at transfer.

| Sparsity At Transfer | Sparsity At Pretraining | | | | | |
| --- | --- | --- | --- | --- | --- | --- |
| | 0.00% | 20.00% | 36.00% | 48.80% | 59.04% | 67.23% |
| 0.00% | 71.59% | 71.89% | 73.44% | 73.70% | 73.63% | 73.52% |
| 20.00% | 71.83% | 72.44% | 73.97% | **74.01%** | 73.52% | 73.39% |
| 36.00% | 71.68% | 72.80% | 73.46% | 73.47% | 72.95% | 72.70% |
| 48.80% | 71.13% | 71.87% | 72.14% | 72.40% | 71.87% | 71.28% |
| 59.04% | 69.26% | 70.31% | 70.61% | 70.73% | 70.11% | 69.38% |

## B.3 MORE ABLATIONS ON IMAGE DATA

**Effects of sparsity at two stages.** We conduct a set of ablation experiments to study the effects of two sparse masks in the Bi-RPT formulation on ResNet-18 (pretrained by ImageNet-C, fine-tuned on CUB-200). We compare against three baselines: fixing $\boldsymbol{m}_s$, fixing $\boldsymbol{m}_t$, and fixing both of them.

Table A7: Test accuracy of fine-tuned ResNet-18 on CUB-200 (10-shot) after pretrained on ImageNet, under different levels of sparsity at pretraining and sparsity at transfer.

| Sparsity At Transfer | Sparsity At Pretraining | | | | | |
|---|---|---|---|---|---|---|
| | 0.00% | 20.00% | 36.00% | 48.80% | 59.04% | 67.23% |
| 0.00% | 38.66% | 35.88% | 38.30% | 39.23% | 40.73% | 39.14% |
| 20.00% | 38.90% | 36.56% | 38.66% | 40.14% | **40.78%** | 39.33% |
| 36.00% | 38.42% | 36.31% | 38.95% | 40.14% | 40.32% | 38.97% |
| 48.80% | 38.13% | 35.23% | 37.80% | 38.02% | 38.37% | 35.69% |
| 59.04% | 37.02% | 33.21% | 35.83% | 35.54% | 35.55% | 32.78% |

Table A8: Test accuracy of fine-tuned ResNet-18 on CUB-200 (10-shot) after pretrained on ImageNet-C, under different levels of sparsity at pretraining and sparsity at transfer.

| Sparsity At Transfer | Sparsity At Pretraining | | | | | |
|---|---|---|---|---|---|---|
| | 0.00% | 20.00% | 36.00% | 48.80% | 59.04% | 67.23% |
| 0.00% | 32.14% | 34.29% | 38.07% | 36.12% | 38.21% | 36.85% |
| 20.00% | 32.53% | 35.99% | 39.63% | 37.92% | **39.94%** | 37.66% |
| 36.00% | 32.52% | 36.07% | 38.99% | 38.56% | 39.07% | 36.95% |
| 48.80% | 31.69% | 34.85% | 37.59% | 36.45% | 36.69% | 34.79% |
| 59.04% | 30.64% | 32.55% | 34.79% | 33.31% | 33.72% | 32.64% |

The performance comparison is shown in Table A9, where we can see that learning masks at both stage yields the highest performance.

**Effects of** $\gamma$   We conduct a set of ablation experiments to study the effects of different $\gamma$ again on ResNet-18 (pretrained by ImageNet-C, fine-tuned on CUB-200).We vary $\gamma$ with in *i.e.,* $\{0.5, 1, 2, 3\} \times 10^{-4}$, and we present the results in Table A10. We show that $1 \times 10^{-4}$ yields the highest performance among all the choices.

**Effects of learning rates.** We conduct a set of ablation experiments on ResNet-18 (pretrained by ImageNet-C, fine-tuned on CUB-200) to study the effects of different learning rate on $\boldsymbol{m}_s$ and $\boldsymbol{m}_t$. The learning rates we study in this ablation experiments are $\{2.5, 3.0, 3.5, 4.0, 4.5, 5.0\}$. We present the test accuracies in Table A11, and we observe that Bi-RPT can stably outperform baselines (74.01%) within a wide range of $\lambda_m$.

Table A9: Ablation study on different sparse masks on image data. "Fixed" means the value of $h$ unchanged. We study the combination of pretraining on ImageNet-C and transferring to Birds.

| Mask Type | Test Accuracy |
|---|---|
| Fixed $\boldsymbol{m}_s$ and $\boldsymbol{m}_t$ | 71.58% |
| Fixed $\boldsymbol{m}_s$ | 72.09% |
| Fixed $\boldsymbol{m}_t$ | 75.53% |
| Ours (Bi-RPT) | 76.29% |

Table A10: Ablation study on the effects of different $\gamma$ on the image data. Test accuracy of fine-tuned ResNet-18 on CUB-200 after pretrained on ImageNet-C is reported.

| $\gamma$ | Test Accuracy |
|---|---|
| $0.5 \times 10^{-4}$ | 72.32% |
| $1 \times 10^{-4}$ | 76.29% |
| $2 \times 10^{-4}$ | 65.42% |
| $3 \times 10^{-4}$ | 52.59% |

Table A11: Ablation study on the effects of different learning rate on $\boldsymbol{m}_s$ and $\boldsymbol{m}_t$ on image data. Test accuracy of fine-tuned ResNet-18 on CUB-200 after pretrained on ImageNet-C is reported.

| $\lambda_m$ | Test Accuracy |
|---|---|
| 2.5 | 72.88% |
| 3.0 | 75.73% |
| 3.5 | 76.29% |
| 4.0 | 75.94% |
| 4.5 | 73.69% |
| 5.0 | 73.34% |

## B.4 VISUALIZATION

We visualize the sparsity pattern learned by Bi-RPT at two stages in Figure A4.

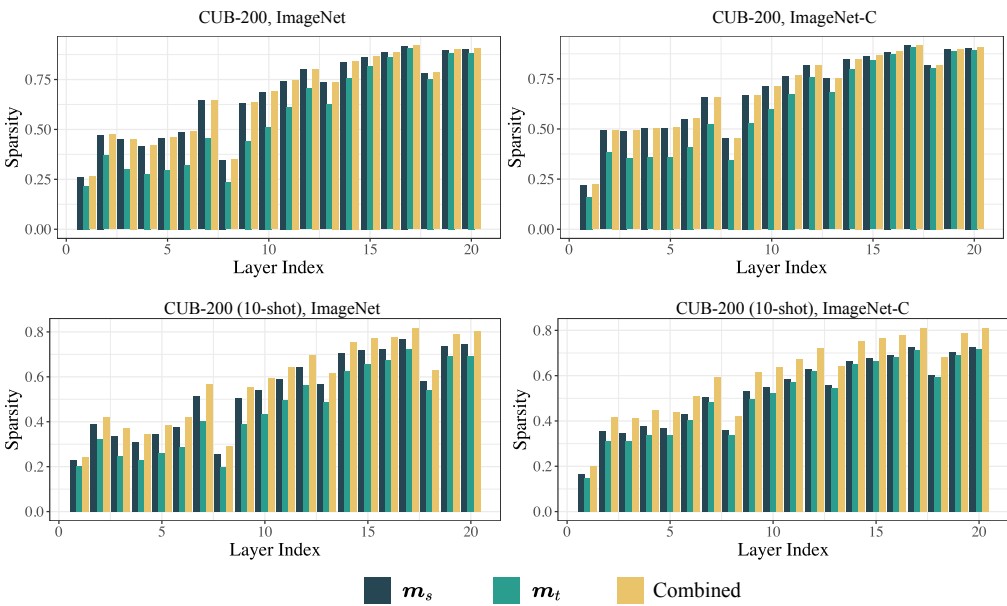

Figure A4: Layerwise sparsity learned by Bi-RPT on CUB-200 and Birds-S with ImageNet and ImageNet-C pretraining. We report the sparsity level of the two masks, as well as their combined sparsity (note that Bi-RPT allows for the two masks to partially overlap).

### B.5 HEA DATA REPRESENTATIONS

The raw inputs for our ML predictor are the alloy's composition and the temperature where the experiment is conducted; therefore, they are 11-dimensional vectors. We map these vectors into 2D images following the pipeline shown in Figure A5. Given a formulation of an alloy, the periodic table representation (PTR) sets the percentage of each element into a specific position according to its position in the periodic table; and the randomized periodic table representation (RPTR) sets the percentage of each element with a pre-defined shuffled periodic table. In our experiments, we use the RPTR to map values in a more balanced way.

## C ADDITIONAL EXPERIMENTS

### C.1 UNCERTAINTY QUANTIFICATION

We provide additional analysis of uncertain quantification. We ensemble ten models trained with Bi-RPT and pretrain-and-transfer (PT) methods by averaging their predictions (Lakshminarayanan et al., 2017), and calculate the standard deviation of the predictions as the uncertainty. The results after ensemble are shown in Table A12.

we show that an ensemble of sparse models provides more reliable results compared to the pretrain-and-transfer baseline. Compared with the ensemble of dense models (PT), the ensemble of sparse models also exhibits strong correlation between the uncertainty and the prediction error.

## D DATASET COMPARISON

We have provided a comparison between different relevant datasets in Table A13. We elaborate more on the differences:

1. Maresca & Curtin (2020) have only sparse data from the Mo-Nb-Ta-V-W element family.
2. Lee et al. (2021a) has released a database of the predicted yield strength of 10 million alloys from the Al-Cr-Mo-Nb-Ta-V-W-Hf-Ti-Zr family at 1300 K. Our dataset contains alloys

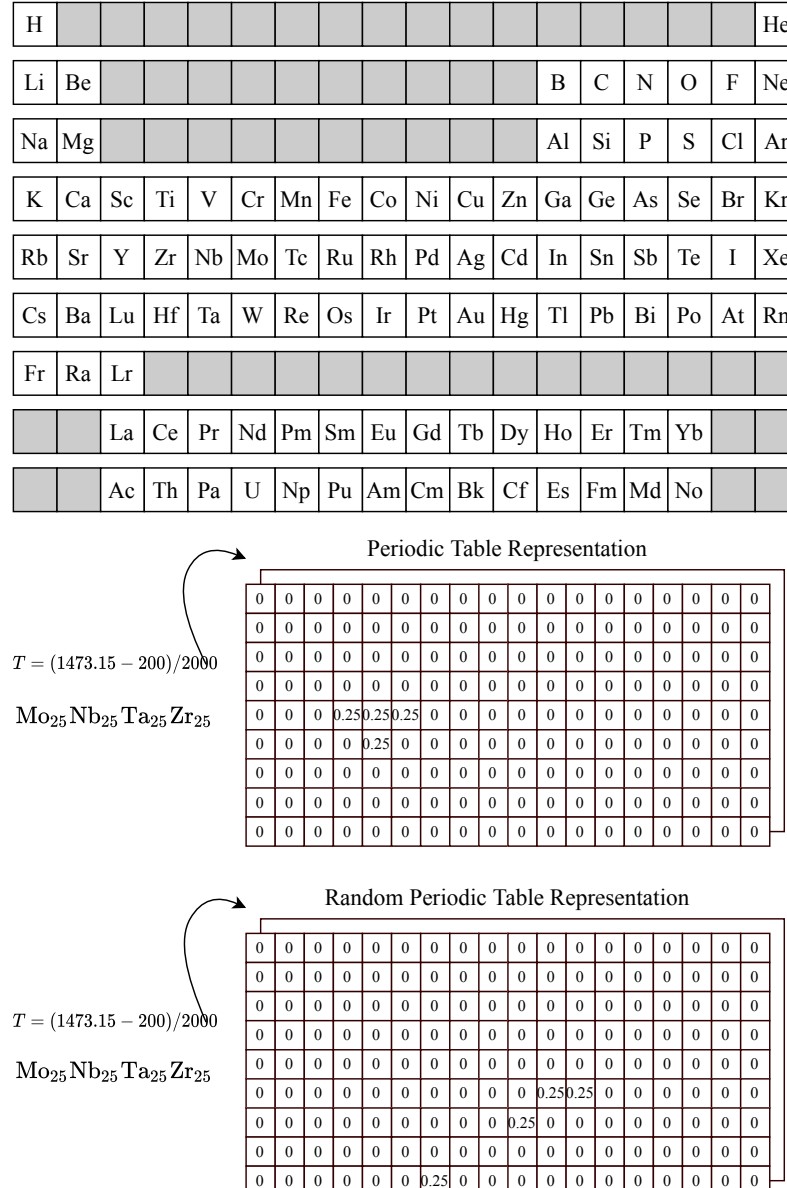

Figure A5: The pipeline for converting a raw input into a pseudoimage. The temperature is embedded as the value of the second channel.

from Al-Cr-**Fe**-Mo-Nb-Ta-V-W-Hf-Ti-Zr family at temperatures from **300 K** to **2500 K**. Our simulation data are significantly larger (over 3 billion samples). The whole simulation data will be available, while only 100K are included for training the ML models in this study.

3. Borg et al. (2020) compiles experimental data from published material science articles since 2004. The dataset contains 630 samples with different crystal structures. Our experimental dataset also compiles experimental data from published material science articles too, but we have also sub-selected the data points using material science domain knowledge. Specifically, we only focus on alloys with BCC structures in contrast to Borg et al. (2020).

Table A12: Uncertainty estimation calculated by ensembling independently trained models. We study two methods: pretrain-and-transfer (PT) and Bi-RPT. The results after ensemble are reported as PT-Ensemble and Bi-RPT-Ensemble, respectively. The estimated uncertainty is reported in brackets.

| Alloys | Temperature (K) | Predicted Yield Stress (GPa) | | | | Experimental (GPa) |
|--------|-----------------|--------|------------------|--------|-------------|---------------------|
| | | Bi-RPT | Bi-RPT-Ensemble | PT | PT-Ensemble | |
| MoNbTaTi | 293.15 | 1.078 | **1.158 (0.083)** | 1.062 | 1.054 (0.011) | 1.210 |
| | 473.15 | 0.965 | 1.046 (0.087) | **0.902** | 0.908 (0.015) | 0.868 |
| | 673.15 | 0.746 | 0.850 (0.085) | **0.731** | 0.740 (0.026) | 0.685 |
| | 873.15 | 0.508 | 0.674 (0.103) | **0.584** | 0.604 (0.021) | 0.593 |
| | 1273.15 | 0.425 | 0.482 (0.088) | 0.488 | **0.501 (0.018)** | 0.539 |
| MoNbTaTiW | 298.15 | **1.268** | **1.268 (0.098)** | 1.068 | 1.062 (0.011) | 1.399 |
| | 873.15 | **0.677** | 0.798 (0.102) | 0.607 | 0.624 (0.022) | 0.689 |
| | 1073.15 | 0.618 | **0.681 (0.111)** | 0.523 | 0.528 (0.013) | 0.674 |
| | 1273.15 | 0.536 | **0.567 (0.124)** | 0.486 | 0.496 (0.017) | 0.620 |
| HfMoNbTaTiZr | 296.15 | **1.527** | 1.392 (0.122) | 1.132 | 1.142 (0.021) | 1.515 |
| | 873.15 | 0.861 | **0.864 (0.098)** | 0.685 | 0.698 (0.017) | 0.973 |
| | 1073.15 | **0.762** | 0.747 (0.105) | 0.612 | 0.624 (0.022) | 0.791 |
| | 1273.15 | **0.662** | 0.646 (0.134) | 0.573 | 0.587 (0.022) | 0.753 |

Table A13: Comparison between different datasets.

| Dataset | Alloy Family | Number of data points | Temperature |
|---------|--------------|-----------------------|-------------|
| Maresca & Curtin (2020) | Mo-Nb-Ta-V-W | Sparse | N/A |
| Lee et al. (2021a) | Al-Cr-Mo-Nb-Ta-V-W-Hf-Ti-Zr | 10 million | 1300 K |
| Borg et al. (2020) | N/A | 630 | N/A |
| X-Yield (Ours) | Al-Cr-Fe-Mo-Nb-Ta-V-W-Hf-Ti-Zr | 3 billion | 300 K - 2500 K |

