# OpenReview forum: "Cross-Quality Few-Shot Transfer for Alloy Yield Strength Prediction: A New Material Science Benchmark and An Integrated Optimization Framework"
_ICLR.cc/2023/Conference — Submitted to ICLR 2023_

### Official Review · Reviewer_THpP · 2022-10-23

**Confidence:** 3
**Correctness:** 3
**Technical Novelty And Significance:** 2
**Empirical Novelty And Significance:** 2
**Recommendation:** 3

**Clarity, Quality, Novelty And Reproducibility:**

The paper has good clarity overall, except the method section (section 4). I had to read several times to get the gist. The problem here includes
1. the use of sparse regularizers are not well motivated: there's recent success on those methods, but why using them here?
2. Too many notations without a clear structure. Some notations, e.g. the dot operator introduced in Equation (1), theta, L_t, R, L_s did not seem to be formally defined. Some of these notations, e.g. theta was introduced before using them in equations.

This section is one of the most important parts in this paper and it needs to be better structured. Try writing a summary paragraph before introducing all these different methods, and always define the notations before using them.

It is difficult for me to assess the quality of the contributions on dataset since I don't have any background on material science. But the two parts of the dataset are already published so maybe that indicates a strong quality. The results of the new method look very good as it consistently outperform baselines. But at the same time, I think the baselines are rather basic here. I think it will be more convincing to at least add an experiment comparing to a multi-fidelity machine learning model, e.g. based on Bayesian linear regression.

As mentioned above, the proposed method does not provide any clarifications on how uncertainty can be predicted in their model, which is a big drawback on the quality of the method.

There is also a significant problem on novelty: the dataset is composed of subsets of simulated data from Maresca & Curtin (2020) and the high-fidelity data from Borg et al. (2020). I don't think the X-Yield dataset can be novel by directly combining two existing dataset. Can the authors clarify what exactly are the work done by them to create this dataset? Claiming it is the first multi-fidelity dataset for HEAs, in my view, touches on a grey area and raises concerns on plagiarism.

Another red flag to me is that the subsets were "carefully filtered" from the original data, raising a question of whether the test data are cherry picked. I hope the authors can clarify more on this issue.


**Details Of Ethics Concerns:**

The open sourced dataset consists of simulated data from Maresca & Curtin (2020) and the high-fidelity data from Borg et al. (2020). Please double-check the copyrights on using these data from existing work.

Despite these sources, the authors claimed their dataset is novel: "While there are existing experimental databases (Borg et al., 2020) and models to predict high-temperature yield strength in HEAs (Maresca & Curtin, 2020), to our best knowledge, this is the first multi-fidelity dataset in the public domain that combines real experimental measurements and large quantities (over 100K) of simulation data for mechanical property prediction in HEAs." In my opinion, this raises potential integrity issues that need to be further reviewed.

**Strength And Weaknesses:**

This work is very well motivated. Discovering new materials is an important task in science and making sure a good predictor exists is a primary sub goal.

To make machine learning methods work on real world tasks is not trivial. The authors have done extensive analyses on the dataset and specialized the method that not only works on their task of interest, but also has the potential to be used for other transfer learning tasks that involves multi-fidelity. Preliminary results were shown on image data.

The main weakness that I see is that the authors only focused on the prediction accuracy, and mentioned nothing about the uncertainty predictions that would be critical to the subsequent experimental design task for new material discovery. I worry that when proceeding to the next step, the authors will have to redo their modeling because of this missing uncertainty prediction component.

Moreover, since discovering new material with high yield strength is the task, I think it wouldn't be too much trouble for this paper to show some preliminary results on simulated new material discovery given their existing dataset. This will also avoid the issue on missing component in the modeling choice in the first place.

Another weakness is the lack of baselines and related work. Closely related works are not discussed, e.g. on multi-fidelity machine learning, domain adaptation and sim to real transfer. The followings are some I found with a quick search.

https://www.sciencedirect.com/science/article/abs/pii/S0927025616306188
https://www.sciencedirect.com/science/article/abs/pii/S0045782517307612
Citations in https://en.wikipedia.org/wiki/Multifidelity_simulation
papers mentioned in https://www.v7labs.com/blog/domain-adaptation-guide
https://link.springer.com/chapter/10.1007/978-3-030-33950-0_25
https://arxiv.org/pdf/2009.13303.pdf

**Summary Of The Paper:**

This paper aimed to discover high-entropy alloys with high yield strength, by first predicting the yield strength. This work presented a dataset called X-Yield for this task and proposed a bi-level optimization method called Bi-RPT as their corss-quality few shot transfer workflow.

**Summary Of The Review:**

This paper brings a very interesting problem in material science to the machine learning community. However, given the technical issues and potential integrity problems, I cannot recommend acceptance at this point but encourage the authors to address those issues in the rebuttal so that it's possible for me to adjust the rating.

---

> ### Author Response · Authors · 2022-11-16
> **Response to Reviewer THpP (1/5)**
>
> We appreciate the reviewer’s efforts in providing valuable feedback and for acknowledging that our work is well-motivated. Responses to your comments are provided below to address your concerns.
>
>
> **[Question 1: Ethics Concerns]**
>
> It seems that the reviewer has fundamental misunderstandings of our dataset contribution. Respectively but very firmly, we disagree with the accusation of plagiarism and do not believe our work raises any integrity issues. This accusation, which surprised us a lot, seems to arise from the reviewer being not quite familiar with how material science data was collected, curated, and filtered. The authors’ institution has already completed the legal review and there are no IPO issues. **Overall, we strongly hope the reviewer to take a careful look at the below response, and consider revising the ungrounded comments and “ethical flags” - we apologize for any confusion caused, but we also take this accusation (that we don’t deserve) seriously and hope to clear your concern sooner.**
>
> **[Simulation Data]** **The simulated data in X-Yield are not pulled from an existing paper or database.** Instead, they were generated by in-house codes developed by ourselves that mainly implement the yield strength computational theoretical model developed in Maresca & Curtin (2020), which we have appropriately cited and discussed in Section 3. We generate data points of alloys from the Al-Cr-Fe-Mo-Nb-Ta-V-W-Hf-Ti-Zr family covering the temperature range from 300K to 2500K, which significantly broadens the availability of simulation data in the literature.
>
> **[Experimental Data]**
> We would like to point out that Borg et al. (2020) have not generated any yield strength data. Instead, they built their **open-source database** based on other already published high entropy alloys (HEA) databases and other research articles, as detailed in their paper. The mined data was placed on an open data platform, Citrination, and is **“intended for use”** as described by the authors. The authors even suggested that “the dataset will be useful as training data for machine learning applications.” Our work has cited their work appropriately.
>
> Our focus is also different from theirs. More specifically, we utilize material domain knowledge to include experimental data that are aligned with our research focus –  RHEAs. More details can be found at the **[Question 3]**.

---

> ### Author Response · Authors · 2022-11-16
> **Response to Reviewer THpP (2/5)**
>
> **[Question 2: The novelty of our dataset]**
>
> Based on the above clarification (in **[Question 1]**), it should be crystal clear that X-Yield is not directly combining two datasets and it is novel in the following aspects:
>
> **Simulation Data Are Novel.** The yield strength values are taken under challenging experimental conditions and each measurement can take between two to four weeks (considering sample preparation). Thus real experimental data are extremely hard to acquire. Therefore, it is challenging for ML models to generalize well to unseen alloys if only trained on limited amounts of experiment data. We, however, provide a massive amount of simulation data to remedy the data scarcity problem. As aforementioned, the simulated data were generated using an in-house code using the yield strength model developed in the publication by Maresca & Curtin (2020). Compared to existing datasets, our simulation data:
>
> - *Contain more compositions of alloys*: We introduce an additional element Fe, known for high strength and moderate ductility, which are important features in designing high entropy alloys.
> - *Cover wider temperature*: We generate yield strength data points in the temperature range from 300 K to 2500 K. As the yield strength of a material is greatly dependent on the temperature, it is very important to have yield strength data points that span a large temperature range.
>
> To fully cover the relevant alloy composition and temperature space we generated over three billion data points of which 100K were randomly selected for inclusion in the machine learning model. This is a significant increase in the number of simulation data points currently available in the literature. We will make all three billion data points available.
>
> **Combination/Problem Setting Is Novel.** X-Yield combines data from both sources and is the first multi-fidelity dataset for the material science field. One of the main characteristics of X-Yield is the quality gaps (data noise, domain gaps, etc.) between the simulation and the experimental data, and learning models under such quality gaps is non-trivial. This leads to a novel yet meaningful problem setting for the material science field. Further, X-Yield reveals the fact that using sparse regularizers can help models better leverage the simulation data and greatly improve the prediction performance, showing a promising and efficient way for the discovery of RHEAs with high yield strength.

---

> ### Author Response · Authors · 2022-11-16
> **Response to Reviewer THpP (3/5)**
>
> **[Question 3: The meaning of “carefully filtered”]**
>
>
> We clarify that **we are not cherry-picking samples** from the original data.
>
> **For the simulation data**, we perform stratified random sampling (from ternary to septenary) from the 3 billion simulation data.
>
> **For the experiment data**, we use our domain knowledge to involve only the alloys with a BCC structure. This is what was meant by “carefully filtered”. Borg et al. (2020) **compiled** data points from existing publications to construct their dataset. The alloys reflected in the database have various crystal structures, such as face-centered cubic (FCC), hexagonal close-packed (HCP), and body-centered cubic (BCC) that are not relevant to our focus -- refractory high entropy alloys (RHEAs). Our interest is in generating a yield strength model for RHEAs with a BCC structure as they are predicted to have a high yield strength at elevated temperatures. The model we used to predict the yield strength in the simulated data points is also specifically for BCC structures. The yield strength is due to edge dislocation motion. Dislocations move through different crystal structures differently, thus one needs to be mindful of crystal structure when selecting data points to create a yield strength model. Thus, we selected only alloys with a BCC structure from the Al-Cr-Fe-Mo-Nb-Ta-V-W-Hf-Ti-Zr element family at temperatures above 300 K to reflect the same design conditions as the simulation data points.

---

> ### Author Response · Authors · 2022-11-16
> **Response to Reviewer THpP (4/5)**
>
> **[Cons 1: Missing uncertainty quantification]**
>
> We have followed your suggestions to analyze the predictive uncertainty. We calculate the standard deviation in predictions due to model ensembling as the metric of uncertainty [r1]. In the table below, we show that an ensemble of sparse models provides more reliable results compared to the pretrain-and-transfer baseline (or dense ensemble). Results are summarized next for ensembles with Bi-RPT and PT in the following format: MSE (uncertainty).
>
> | Alloy | Temperature | Bi-RPT| PT | Bi-RPT-Ensemble | PT-Ensemble |  Experimental (GPa) |
> | :---: | :---: | :---: | :---: | ---: | :---: | :--: |
> | MoNbTaTi | 293.15 | 1.078 | 1.062 | 1.158 (0.083) | 1.054 (0.011) | 1.210|
> | MoNbTaTi | 473.15 | 0.965 | 0.902 | 1.046 (0.087) | 0.908 (0.015) | 0.868 |
> | MoNbTaTi | 673.15 | 0.746 | 0.731 | 0.850 (0.085) | 0.740 (0.026)|  0.685 |
> | MoNbTaTi | 873.15 | 0.508 | 0.584 | 0.674 (0.103) | 0.604 (0.021) | 0.593 |
> | MoNbTaTi | 1273.15 | 0.425 | 0.499 | 0.482 (0.088) | 0.501 (0.018) | 0.539 |
> | MoNbTaTiW | 296.15 | 1.268 | 1.068 |  1.268 (0.098)| 1.062 (0.011) | 1.399 |
> | MoNbTaTiW | 873.15 | 0.677 | 0.607 | 0.798 (0.102) | 0.624 (0.022 )| 0.689 |
> | MoNbTaTiW | 1073.15 | 0.618 | 0.523 | 0.681 (0.111)| 0.528 (0.013) |  0.674 |
> | MoNbTaTiW | 1273.15 | 0.536 | 0.486 | 0.567 (0.124) | 0.496 (0.017) | 0.620 |
> | HfMoNbTaTiZr | 296.15 | 1.527 | 1.132 | 1.392 (0.122)  | 1.142 (0.021)  | 1.515 |
> | HfMoNbTaTiZr | 873.15 | 0.861 | 0.685 | 0.864 (0.098) | 0.698 (0.017) | 0.973 |
> | HfMoNbTaTiZr | 1073.15 | 0.762 | 0.612 | 0.747  (0.105) | 0.624 (0.022) |  0.791 |
> | HfMoNbTaTiZr | 1273.15 | 0.662 | 0.573 | 0.646	 (0.134) | 0.587 (0.022) | 0.753 |
>
> The results suggest that the pretrain-and-transfer method generates over-confident predictions, as the standard deviations are small but the ground truth values are not in the confidence intervals. The estimated standard deviation of the prediction of Bi-RPT is larger, but the corresponding confidence interval covers the true values.
>
> **[Cons 2: Simulation on new material discovery]**
>
> It is extremely non-trivial to experimentally measure the yield strength of a new alloy due to the complex process of sample preparation and experiments. To fit in the rebuttal time period, we set up a different task to see the quality of our models’ recommendations. We train our model on 10% of experimental data, and we predict the yield strength of the remaining 90% of experimental data. We assess the quality of the prediction by calculating the  **Spearman correlation** between the predicted and the experimentally measured yield strength. The higher the correlation is, the more accurate the model can produce the correct ranks of alloys’ strength. We also provide the Spearman correlation for data points at elevated temperatures, which is closer to our design goal.
>
> | Method | Spearman Correlation | Spearman Correlation (>800K) |
> | :-------: | :-------: |  :--: |
> | Bi-RPT | 0.875 | 0.589 |
> | PT | 0.839 | 0.335 |
> | NP | 0.840 | 0.359 |
>
> We can observe that our method (Bi-RPT) has the highest correlation between the prediction and the true values with all the testing samples, and the performance gap becomes even more significant when we are only focusing on high temperatures (>800K).
>
> We then zoom in to look at the prediction results at one certain temperature (1073.15K). The top 5 candidate alloys generated by Bi-RPT, PT, NP, as well as the top 5 candidates in the unseen (i.e., testing) experimental samples are shown in the table below.
>
> | Rank | Bi-RPT | PT | NP | Experimental |
> | :--: | :--: | :--: | :--: | :--: |
> | 1 | CrMoNbTi | Nb$\_{0.2}$Ti$\_{0.2}$V$\_{0.4}$Zr$\_{0.2}$ | Ti$\_{0.2}$V$\_{0.4}$W$\_{0.2}$Zr$\_{0.2}$ | CrMoNbTi |
> | 2 | AlCrMoNbTi | Al$\_{0.22}$Cr$\_{0.11}$Nb$\_{0.22}$Ti$\_{0.22}$V$\_{0.22}$ | CrMoNbTi | HfMoTaTiZr |
> | 3 | HfMoTaTi | AlNbTiV | MoNbTaVW | HfMoNbTaTiZr |
> | 4 | HfMoNbTaTiZr | MoNbTaVW | Al$\_{0.22}$Cr$\_{0.11}$Nb$\_{0.22}$Ti$\_{0.22}$V$\_{0.22}$ | HfMoNbTaZr |
> | 5 | Ti$\_{0.2}$V$\_{0.4}$W$\_{0.2}$Zr$\_{0.2}$ | MoNbTaTiVW | MoNbTaTiVW | AlCrMoNbTi |
>
> Bi-RPT correctly predicts three alloys that have the largest five yield strength values at 1073.15K. The other two baseline methods, however, fail to identify any top five alloys at this specific temperature. This exhibits the potential of Bi-RPT on identifying novel alloys that have high yield strengths.
>
> [r1] Simple and Scalable Predictive Uncertainty Estimation using Deep Ensembles

---

> ### Author Response · Authors · 2022-11-16
> **Response to Reviewer THpP (5/5)**
>
> **[Question 4: the motivation of using sparsity]**
>
> The motivation of using sparsity is clearly established. We have explained in paper (summarized in the paragraph below figure 2; more supporting references in section 2.2). We hereby explain again for you.
> 1. Sparsity as a regularizer in deep learning has been separately studied to solve data noise & domain gap [r2,r3,r4]; and to solve data scarcity [r5,r6]. Since both challenges are truly presented simultaneously in our real world material science application, we are naturally **motivated** to look at model sparsification as the unified tool.
> 2. However, since we now need to tackle two challenges together in the two stages (pre-training/transfer) pipeline, with two goals (noise robustness/data efficiency), how to allocate sparsity (levels, and mask locations) optimally in two stages becomes a nontrivial decision-making, and infeasible to be manually solved. Our **novelty** on sparsity is hence to formulate the bi-level sparse optimization form Bi-RPT, that leads to a novel principled way to automatically decide sparsity levels and masks. Our extensive experiments (both image, and material science) thoroughly show Bi-RPT to outperform vanilla stage-wise tuning.
>
> [r2] The Generalization-Stability Tradeoff In Neural Network Pruning
>
> [r3] Understanding the effect of sparsity on neural networks robustness
>
> [r4] Sparse DNNs with improved adversarial robustness.
>
> [r5] Data-efficient gan training beyond (just) augmentations: A lottery ticket perspective.
>
> [r6] An embarrassingly simple baseline to one-shot learning
>
>
> **[Cons 3: Notations and Summary of the methods]**
> Thank you for the suggestions. We have revised the notions and the equations for better presentation in the revision. A brief paragraph has also been added to summarize the methods.
>
>
> **[Cons 4:  Comparison against baselines Bayesian linear regression]**
>
> We have followed your suggestion to compare against a recent Bayesian regression method [r7]. The results are shown below where we can see Bi-RPT still outperforms the Bayesian regression baseline:
>
> |Method | Test MSE (10% training data) |
> | :--: | :--: |
> | Bi-RPT | 0.092 |
> | Bayesian  | 0.100 |
>
> We have further compared with other baseline methods: (1) a cross-domain few-shot learning method AFA [r8], (2) a noisy label learning algorithm [r9]. The results are collected in the table below:
>
> | Method | Testing Accuracy (10% training data) |
> | :--: | :--: |
> | Bi-RPT (ours) | 66.83% |
> | AFA [r8] | 57.36% |
> | Universal Probabilistic Model [r9] | 54.38% |
>
>
> [r7] Multi-fidelity regression using artificial neural networks: efficient approximation of parameter-dependent output quantities
>
> [r8] Adversarial Feature Augmentation for Cross-domain Few-shot Classification
>
> [r9] Tackling Instance-Dependent Label Noise via a Universal Probabilistic Model

---

> ### Author Response · Authors · 2022-11-18
> **Response to Reviewer THpP**
>
> Dear Reviewer THpP:
>
> Thank you so much for your time and effort in reviewing! For every concern in your comments, we have provided a detailed response. Would you mind checking them to see if they have successfully addressed your questions and concerns? If you have other questions or you require additional clarification, please let us know soon so that we can provide more responses before the author-reviewer discussion period ends.
>
> Best Regards,
> Authors

---

> ### Author Response · Authors · 2022-11-23
> **We are looking forward to discussing with you**
>
> Dear Reviewer THpP,
>
> We thank you again for your time and comments. We believe that there was a major misunderstanding, and we are eager to initiate a discussion with you.
>
> In our previous responses, we have made clarifications on our dataset and provided point-wise explanations to your other questions. Specifically, we compared against the Bayesian regression method, provided uncertainty quantification results, performed a simulated material discovery experiment, and clarified again the motivation of our methods. We genuinely hope you could check our response and see if we have addressed all the concerns. Thank you very much in advance!
>
> Best wishes,
>
> Authors

---

> ### Author Response · Authors · 2022-11-28
> **To Reviewer THpP**
>
> Please kindly check our newest public letter to AC/SAC and all reviewers. We treat your accusation as a very serious matter and want to make every possible effort to clarify your misunderstanding. We are eagerly waiting for your feedback.
>
> As we stated in the public letter, due to the very serious nature of the accusation made, and the well-known public record nature of OpenReview forums, we simply cannot let this ungrounded and factually mistaken accusation regarding our work integrity be left here publicly and permanently.
>
> Thanks for your time!

---

> > ### Comment · Reviewer_THpP · 2022-11-28
> > **Thank you for the rebuttal**
> >
> > Really appreciate the authors' detailed responses.
> >
> > "...the simulated data were generated using an in-house code using the yield strength model developed in the publication by Maresca & Curtin (2020)"
> > "The mined data was placed on an open data platform, Citrination, and is “intended for use” as described by the authors."
> > "For the simulation data, we perform stratified random sampling (from ternary to septenary) from the 3 billion simulation data.
> >
> > For the experiment data, we use our domain knowledge to involve only the alloys with a BCC structure. This is what was meant by “carefully filtered”. Borg et al. (2020) compiled data points from existing publications to construct their dataset."
> > "both Borg et al. (2020) and we collected data from published material science articles "
> >
> > It is still unclear to me why simulation data needs to be subsampled given that they are generated by the authors. It is also somewhat concerning how the data was collected from published articles that potentially have been partly patented. Please note that I'm not accusing the authors of plagiarism at all. Unfortunately, given the unclear descriptions and the stark differences between ML and material science, I believed it was necessary for an expert in this particular area to examine potential issues regarding copyrights etc. As an expert on ML with no material science background, I still cannot tell what exactly this dataset is trying to accomplish (BCC? FCC? RHEAs?) or how exactly it was collected after reading the rebuttal. In fact, why do the authors prefer to publish an important material science dataset at a ML conference? I think the authors can benefit greatly from separating the ML algorithm in this work from the dataset and just submit the paper on their new algorithm to ML conferences.
> >
> > Again, I really appreciate that the authors took this matter very seriously and wrote several very detailed responses. Please refer to the AC and Ethics Committee for follow-ups as I am intrinsically unable to objectively examine the legal issues and I don't know what the procedure is. As a reviewer, the only responsibility I'm taking is flagging ***potential*** issues, which, again, I'm not claiming the issue exists or not. Since the authors' institution has already performed legal reviews, perhaps they are aware of how to address such problems under a double-blind reviewing situation?

---

> > > ### Author Response · Authors · 2022-11-29
> > > **Further Response to Reviewer THpP**
> > >
> > > Thank you for your additional comments. We further provide answers to your questions:
> > >
> > > **Question 1: Why does the simulation data need to be subsampled?**
> > >
> > > The total amount of simulation data we have generated is around **3 billion**. We work on a subsampled set in this paper, to avoid the prohibitive computational burden of training over the full set. However, as promised, we will make the whole 3 billion simulation data publicly available.
> > >
> > > **Question 2: What does the dataset aim to achieve?**
> > >
> > > The work aims at facilitating machine learning methods for HEA yield strength prediction. We focus on an important subclass of HEA, i.e. refractory HEA, which has been shown to keep mechanical properties at elevated temperatures. Further, BCC and FCC are two different crystal structures for HEAs. We are focusing on BCC-structured refractory HEA, which is believed to have high yield strengths.
> > >
> > > **Question 3: How the data were collected?**
> > >
> > > **Simulation data**: As we discussed in Section 3, the simulation data were generated by in-house codes developed by ourselves that mainly implement the yield strength computational theoretical model developed by Maresca & Curtin (2020).
> > >
> > > **Experimental data**: We manually collect alloys with BCC structures from Citrination with our domain knowledge. Note that the data points are all publicly available both in their original published articles and Citrination. For example, one can find the yield strength for 6 different alloys containing Aluminum in [r1] and in Citrination. The rest of the data points are collected by the same routines.
> > >
> > > We would be happy to clarify further if the reviewer has specific questions or confusion.
> > >
> > > [r1] Microstructure and Properties of Aluminum-Containing Refractory High-Entropy Alloys
> > >
> > > **Question 4: Why publish at ML conference?**
> > >
> > > We indeed expect our dataset to generate impacts in both material science and ML fields, and we chose ICLR as the target venue because:
> > >
> > > (i) our work represents a new problem presented for the current exciting ML initiative of “AI4Science”, that ICLR explicitly encourages. Currently, ML for material science remains a niche area, often due to a lack of standardized data (most ML researchers unfortunately don’t read material science journals). We hope our paper if published in ICLR, would stimulate more ML researchers to work on a well-defined standard benchmark, whose success would make direct and highly meaningful contributions to reduce scientific experiment costs.
> > >
> > > (ii) the problem, even abstracted from the material science context to a pure ML setting, constitutes a highly non-trivial “synthetic-to-real” problem with ML research merits. It involves both challenges of large domain gap and few-shot transfer. Syn2real algorithms are often studied in CV and speech fields. Our material science benchmark provides a new testbed for those algorithms and motivates new ones.
> > >
> > > Therefore, we think our paper suits the ICLR scope nicely and would benefit both fields (material science and ML) more if published at this venue.
> > >
> > >
> > > **Ethic Flag**
> > >
> > > Thanks for your suggestion We have already reached out to PCs and the ethics committee to handle this issue separately. Updates will be shared with you as soon as available: we’re extremely confident no ethical concern will be found eventually.
> > >
> > >
> > > From what we understood, we have fully addressed all the technical concerns you raised. **We sincerely hope you would please reconsider your score, to one that is deserved by a serious interdisciplinary ML work like this piece**: please kindly take into account all the above clarifications, as well as (hopefully soon-coming) ethic flag resolution. If any other concerns are holding you back from increasing your score, we would be happy to address them.

---

### Official Review · Reviewer_NW7Q · 2022-10-25

**Confidence:** 4
**Correctness:** 3
**Technical Novelty And Significance:** 3
**Empirical Novelty And Significance:** 3
**Recommendation:** 6

**Clarity, Quality, Novelty And Reproducibility:**

This paper is well-written and easy to follow. Some techniques have been widely used in other applications, which limits the novelty.

**Strength And Weaknesses:**

Strength:
1. This work presents a large-scale material science benchmark.
2. This work presents a two-stage method to address the scarcity of experimental annotations and the quality gap in the imperfectly simulated data.


Weaknesses:
1. The simulated data would have much noise because of the gap between real and theoretical scenarios. How did you address this problem? Although the authors give a sparsity solution, it lacks theoretical analysis to explain its effectiveness.
2. How did you tune the accurate sparsity in the paper? From tables A5-6, one could find that some sparsity rates are so specific.
3. It is better to discuss this work with noisy labels. Why did you utilize the techniques of learning with noisy labels to address the noise?
4. The utilized techniques are not originally proposed by the authors. These techniques have been widely used in other applications, which limits the novelty.

**Summary Of The Paper:**

In this paper, the authors proposed a large-scale material science benchmark with 240 experimental measurements and over 100K simulated high-entropy alloy yield strength annotations. To address the scarcity of experimental annotations and the quality gap in the imperfectly simulated data, the authors present a noise-robust feature learning regularizer at the pre-training stage, and as a data-efficient learning regularizer at the few-shot transfer stage.

**Summary Of The Review:**

This work is interesting. However, some techniques have been widely used in other applications, which limits the novelty.

---

> ### Author Response · Authors · 2022-11-16
> **Response to Reviewer NW7Q (1/2)**
>
> We thank the reviewer for the valuable feedback on our work. We have provided answers to your questions below.
>
> **[Question 1: How to deal with the noise in data? Theoretical results? ]**
>
> As we mentioned in Section 4, we leverage sparse regularizers to improve models’ robustness to noise in data. Sparsity has been a well-known regularizer for robustness, in both classical and deep learning regimes.  Besides the material science application, we also empirically prove the concept in our image classification experiments where sparsity is extensively tested on data with controlled noise, and sparsity is shown to boost generalization in all apple-to-apple comparison settings.
>
> Similar observations on the effectiveness of sparsity on noise resilience/robust generalization have been vastly found under many learning-under-noise scenarios [r1, r2, r3]. It has also come to our attention that a concurrent ICLR submission [r4] coincidentally provides the first proof that sparsity indeed enhances the generalization of neural networks when input is corrupted by noise when the pruning ratio lies within a certain range (whose lower/upper bounds depend on input SNR): that theoretically solidifies our work further. We will happily cite this work in our final version.
>
> **[Question 2: How to tune the sparsity level?]**
>
> The reviewer was referring to the ”hand-tune” baseline (Tables A5, A6 in Appendix), NOT our main method Bi-RPT. In fact, one main advantage of Bi-RPT is that we do not need to manually tune the sparsity levels. We clarify more below.
>
> For the Hand-Tune baseline, we conduct experiments with numbers of pruning rounds at both stages from 0 to 5 (mentioned in Section B.2). We exhaustively search over all the combinations of pruning rounds, and select the one with the highest downstream task performance (i.e., test accuracy on CUB-200/CUB-200 (10-shot).
>
> For our main method Bi-RPT, we no longer manually tune the sparsity levels. Instead, the model learns the sparse masks in a principled way with our Bi-RPT framework, as mentioned in Section 4. The sparse levels of the two stages are automatically allocated given one overall sparsity constraint, and the experiment results show that the learned sparse patterns have better quality (refer to Table 1 on image results, and Tables 2-3 for material results).
>
> [r1] The Generalization-Stability Tradeoff In Neural Network Pruning
>
> [r2] Understanding the effect of sparsity on neural networks robustness
>
> [r3] Sparse DNNs with improved adversarial robustness.
>
> [r4] https://openreview.net/forum?id=dn6_PK73hAY

---

> ### Author Response · Authors · 2022-11-16
> **Response to Reviewer NW7Q (2/2)**
>
> **[Question 3: Why not use learning with noisy labels?]**
>
> We follow your suggestion and compare it with one recent noisy label learning method [r5]. Following their pipeline, we merge both the low- and high-fidelity data to train the network. Note that 10% of the experimental data and all the low-fidelity experimental data are used as the training samples, and the rest 90% of the experiment data are used for testing. We collect the results in the table below:
>
> | Method | Test Accuracy (5% training samples) |  Test Accuracy (10% training samples) |
> | :------: | :------: | :---: |
> | Universal Probabilistic Model [r5] | 53.71% | 54.38% |
> | Ours (Bi-RPT) | 64.37% | 66.83% |
>
> We observe the method introduced in [r5] shows lower testing accuracy with accuracy gaps of more than 10% under the two settings. The reason is probably that we are dealing with data noises and domain gaps, so narrowing down to only label noise leads to only suboptimal performance.
>
> We are definitely willing to discuss the usage of noisy label learning methods. The reason why we focus on sparsity is that we see sparsity as a unified tool for both challenges in the material science field, i.e, the domain gap & data noise, and data scarcity. Meanwhile, learning with noisy labels is more studied under the classification settings, and it seems not straightforward to extend it to the regime of regression.
>
> **[Question 4: Novelty of our method?]**
>
> We politely yet firmly disagree with your assessment. The authors are quite familiar with the sparsity + bi-level optimization fields, and we understand sparsity is not any new term for ML - neither did we claim so. But the principled way we used the sparsity “twice” to kill two birds in one stone IS NOVEL - as clearly supported by our superior performance over existing methods in the paper.
>
> The novelty of our method lies in two aspects:
> - *Tackling the two challenges at the same time*: While sparsity has succeeded in addressing the challenges of data scarcity and the data noise & domain gap, existing methods often *address one problem at a time*. However, Bi-RPT is a novel solution to *deal with both challenges*, which are truly presented simultaneously in our real-world material science application, using model sparsification as the unified tool.
> - *Principled way to locate sparse masks and levels*: Pruning via the bi-level optimization has not been well studied, and allocating sparsity optimally towards two stages (pre-training/transfer) with two goals (noise robustness/data efficiency) is infeasible to resolve manually. Following this stream of research, Bi-RPT offers a novel principled way to decide sparsity levels and masks. By solving the bi-level optimization problem, Bi-RPT automatically decides the sparse patterns, which are shown to have superior performance on image and real-world material scientific tasks.
>
> [r5] Tackling Instance-Dependent Label Noise via a Universal Probabilistic Model

---

> ### Author Response · Authors · 2022-11-18
> **Response to Reviewer NW7Q**
>
> Dear Reviewer NW7Q:
>
> Thank you so much for your time and effort in reviewing! For every concern in your comments, we have provided a detailed response. Would you mind checking them to see if they have successfully addressed your questions and concerns? If you have other questions or you require additional clarification, please let us know soon so that we can provide more responses before the author-reviewer discussion period ends.
>
> Best Regards,
> Authors

---

> ### Author Response · Authors · 2022-11-23
> **We are looking forward to discussing with you**
>
> Dear Reviewer NW7Q,
>
> We thank you again for your time and comments. As the time windows are closing, we hope we discuss this with you further to see if our answers have addressed your concerns. If there are any further questions, we can provide further answers to them.
>
> Specifically, we have: (1) provided additional experiments with noisy label training and (2) clarified the novelty of our methods. We genuinely hope you could check our response and see if we have addressed all the concerns. Thank you very much in advance!
>
> Best Wishes,
>
> Authors

---

### Official Review · Reviewer_iZzs · 2022-10-28

**Confidence:** 3
**Correctness:** 3
**Technical Novelty And Significance:** 2
**Empirical Novelty And Significance:** 3
**Recommendation:** 6

**Clarity, Quality, Novelty And Reproducibility:**

- Clarity : The paper is well written. Motivations are clear.
- Quality : Selected approaches and baselines are well analysed. But, the paper missed out on other potential baselines common in the ML community.
- Novelty : Borderline : The paper proposes a new dataset for use by the ML for material science community. The synthetic component is simulated. While reading it seems that the real component comes from an existing dataset. The approach used to train models is motivated from an existing one.
- Reproducibility :  The authors provide a code


**Strength And Weaknesses:**

Strengths:
- I like the problem setup, the applications of the proposed dataset and the approach are well motivated.
- The paper proposes a new benchmark dataset for enabling machine learning research for HEA yield strength prediction.
- I like the connections that the paper draws to few shot learning in the ML for science domain.
- The paper proposes a technique for few-shot + domain transfer which is based on sparsity regularization.
- The approach is fairly simple and potentially applicable elsewhere. The authors do verify that it works well compared to some simple baselines on a image based few-shot transfer task.
- The experimental setup is sound and the proposed approach brings about improvements for the task at hand

Weaknesses:
- Baselines : The problem setup falls in the ML research areas of few-shot learning/cross-domain few shot learning, (supervised) domain adaptation/transfer learning. For example, refer to [a,b] These areas have been well studied and are very active areas of research. It is not clear why this particular approach was chosen, and how the approach compares to other popular approaches in these areas.
- Section 3: It would help to add a table comparing the kinds of datasets that exist to supplement the Section 3.
- Section 3: I found the construction of the real part of the dataset (in Dataset Construction) seems misleading - Till that point I was under the impression that both real and synthetic parts were contributions, but according to that paragraph the real part was curated from an existing one. Please explain how your subset is different. In my opinion the contributions should be adjusted as well.
- What happens if you use the weight masking strategy without a two-stage approach ? How does that compare to the variants presented in Table 1 ?
- Question : Given that only 11 elements were used, why encode this as an image - which is mostly sparse?
- Question : Are all configurations of the selected elements realistic. What happens if the model is given an unrealistic combination ?
- Suggestion : Please move "Data representation" above "Architectures and Baselines". I was not expecting a CNN-based model and it was not clear to my why that was used till I got to Data Representations.
- Question : Fig. 3 : Is the drop in relative performance for 1273K due to less data in this regime ?



[a] Guo, Yunhui, et al. "A broader study of cross-domain few-shot learning." ECCV 2020
[b] Zhuang, Fuzhen, et al. "A comprehensive survey on transfer learning." Proceedings of the IEEE



**Summary Of The Paper:**

- The paper targets determining yield strength for high entropy alloys which has numerous practical applications
- The first contribution of the paper is to organize a large-scale material science dataset termed X-Yield. The dataset is composed of  >100K synthetic data points and 240 experimentally measured real data points.
- Since there is a domain gap between the real and synthetic data, the authors propose a technique based on sparsity regularization to learn from source while adapting the model to the target data.

**Summary Of The Review:**

The paper proposes a new benchmark dataset which might be useful for the material science community. The paper is well written but I have a few concerns regarding baselines and novelty of the dataset. My initial rating is 6.

---

> ### Author Response · Authors · 2022-11-16
> **Response to Reviewer iZzs (1/3)**
>
> We sincerely thank the reviewer for appreciating our proposed benchmark and methods. We have provided responses to your comments below.
>
> **[Weakness 1: More baselines?]**
>
> We conduct additional experiments of the classification tasks on X-Yield to compare with a state-of-the-art few-shot method [r1]. We use all the classes during training, and the same classes for testing. To conduct a fair comparison, we follow a standard 5-shot scheme: the model is trained on the low fidelity data, fine-tuned on five samples from each class in the experimental data, then evaluated on the rest unseen experimental data. The mean and standard deviation of the testing accuracy are both reported in the following table.
>
> | Method | Testing Accuracy (10% training data) |
> | :--: | :--: |
> | Bi-RPT | 66.83% (1.41%) |
> | AFA [r1] | 57.36% (0.26%) |
>
> Simply applying their method cannot achieve superior accuracy on X-Yield: Bi-RPT outperforms the few-shot learning method by approximately 9.5%. It is also noteworthy that it is not straightforward to adapt their method for regression tasks, which is essential for discovering new alloys.
>
> **[Weakness 2: Comparison with existing datasets]**
>
> We have followed your suggestion to add a table in Appendix to compare X-Yield with several existing datasets in the material science field.
>
> Three most relevant dataset are:
> - [r2] has only sparse data from the Mo-Nb-Ta-V-W element family.
> - [r3] has released a database of the predicted yield strength of 10 million alloys from the Al-Cr-Mo-Nb-Ta-V-W-Hf-Ti-Zr family at 1300 K. Our dataset contains alloys from Al-Cr-**Fe**-Mo-Nb-Ta-V-W-Hf-Ti-Zr family at temperatures from **300 K to 2500 K**. Our simulation data are significantly larger (over 3 billion samples). **We will make the whole simulation data available**, while only 100K are included for training the ML models in this study.
> - [r4] compiles experimental data from published material science articles since 2004. The dataset contains 630 samples with different crystal structures. Our experimental dataset also compiles experimental data from published material science articles too, but we have also sub-selected the data points using material science domain knowledge. Specifically, we only focus on alloys with BCC structures in contrast to [r4], which we will explain more in the next response.
>
> [r1] Adversarial Feature Augmentation for Cross-domain Few-shot Classification
>
> [r2] Mechanistic origin of high retained strength in refractory BCC high entropy alloys up to 1900K
>
> [r3] Strength can be controlled by edge dislocations in refractory high-entropy alloys
>
> [r4] Expanded dataset of mechanical properties and observed phases of multi-principal element alloys

---

> ### Author Response · Authors · 2022-11-16
> **Response to Reviewer iZzs (2/3)**
>
> **[Weakness 3: Dataset Contribution]**
>
> The major difference between the experiment data in X-Yield and Borg et al. (2020) is the crystal structure. Our goal is to build a yield strength prediction model for refractory high entropy alloys (RHEAs) with a BCC structure. RHEAs with a BCC structure have been predicted to have high yield strength at elevated temperatures [r5].
> However, as Borg et al. (2020) directly compiled 630 different HEAs from existing publications, various non-relevant crystal structures such as face-centered cubic (FCC), hexagonal close-packed (HCP), and body-centered cubic (BCC) are presented. Dislocations, the originality of the yield strength, move through different crystal structures differently; thus the crystal structure needs to be specified when selecting data points to create a yield strength model. To reflect the same design conditions, we further apply the material science domain knowledge to include only the alloys with BCC structures in our experimental data.
>
> We believe the contribution of our dataset is appropriate:
> - *New Simulation data*: We have provided a new collection of simulation data covering alloys of the Al-Cr-Fe-Mo-Nb-Ta-V-W-Hf-Ti-Zr family and a wide temperature range (300K - 2500K), resulting in over three billion data points (of which 100K were randomly selected to train our models). We have greatly increased the number of simulation data currently available in the literature, and **we will make the entire 3 billion samples public**.
>
> - *Experimental data*: We go a step further, using the material science domain knowledge to include the alloys with only BCC crystal structures. These 240 data entries cover nearly all of the existing yield strength measurements on RHEAs with a BCC structure, which can be regarded as one of the most comprehensive benchmarks for the RHEAs discovery (in BCC structures).  Note that in material science, experimental data generation is almost impossible by a single team. Only a total of 240 samples are currently known to the community that too characterized mostly at room temperature -- that is the precise problem.
>
> - *Novel Problem Setting in Material Science*: X-Yield is constructed through a careful selection and combination of experimental and simulation data. Such a combination for the material science field is novel and meaningful: the inclusion of simulation data improves the models’ predictive performance on unseen experimental data, which shows a promising and efficient way for the discovery of RHEAs with a high yield strength (given their scarcity). No prior study has been conducted on this topic to our best knowledge, and our experiments clearly demonstrate that simulated data pre-training improves experimental data generalization, especially under our novel optimization framework.
>
> **[Question 1: Applying sparsity without a two-stage pipeline?]**
>
> We experiment on X-Yield where we train the model with a single-stage pipeline.
>
> We merge the simulation and the experimental data together and learn only one sparse mask for the model. The results are shown in the table below:
>
> | Method | Test Accuracy | Test MSE |
> | :-----: | :-------------: | :------------:|
> | Pretrain-and-transfer| 62.84% $\pm$ 1.96% | 0.162 $\pm$ 0.002 |
> | Single-stage sparsity | 56.76% $\pm$ 2.57%| 0.103 $\pm$ 0.009 |
> | Ours | 64.37% $\pm$ 1.10% | 0.130 $\pm$ 0.006 |
>
> The results in the table suggest that using a single-stage pipeline leads to inferior performance, which advocates the usage of our two-stage design.
>
>
> [r5] Recent Progress with BCC-Structured High-Entropy Alloys

---

> ### Author Response · Authors · 2022-11-16
> **Response to Reviewer iZzs (3/3)**
>
> **[Question 2: Why use pseudo-images to encode alloys?]**
>
> We follow a standard setting in [r6] to adopt the randomized periodic table representation (randomized PTR) because it demonstrates superior performance compared to the simple 1-D representation. Specifically, in [r6] the randomized PTR representation outperforms the 1-D counterpart by 6.6% and 5.0% in terms of training and testing accuracy, respectively.
>
> To further convince the reviewer, we conduct an experiment with 1-D representation and MLP on X-Yield. The MLP has six layers. The input dimension for each layer is 11,64,128,256,128,64, respectively, and the output dimension for each layer is 64,128,256,128,64,5, respectively. We follow the same pipeline to train the MLP directly with 5% of experiment data, and we obtain a testing accuracy of  37.86% and a testing MSE of 0.23. It shows that the 1-D representation (and MLPs) may not lead to optimal performance in our case.
>
>
> **[Question 3: Are configuration realistic?]**
>
> We are not entirely sure what ``unrealistic’’ means here. If the reviewer is referring to alloys that do not exist, then we never gave the model an unrealistic combination as training or testing data input. All the test cases (experimental data) we chose are 100% BCC and realistic.
>
> If an unseen combination is given to the model, then the model will still predict a yield strength for it. Every ML models’ output can have a high error, and so do ours. Although it is not the focus of our paper, there are potential ways to deal with this problem:
> - From the material science side, we can utilize CALPHAD to make sure we only predict the yield strength of only realistic combinations.
> - From the ML side, we could incorporate uncertainty quantification techniques to identify out-of-distribution samples (i.e., unrealistic combinations here).
>
> **[Question 4: The reason for the drop in relative performance]**
>
> The number of data points with a temperature of 1273.15K is 29, which is larger than the number of samples with a temperature of 1473.15K (19 samples) and 1673.15K (4 samples). Therefore, the reason for the drop in relative performance is not due to fewer data under this temperature.
>
> However, we point out that Bi-RPT is performing very similarly compared to other baselines. Specifically, the MSE for Bi-RPT is 0.086, while for PT is 0.081. The gap between the two methods is not significant. We are interested as well in understanding why our method only performs relatively inferiorly at this particular temperature. However, we could not provide a precise answer in the current scope of our work.
>
> **[Suggestions]**
>
> Thank you. We have followed your suggestions to adjust the ordering in the revision.
>
> [r6] A general and transferable deep learning framework for predicting phase formation in materials, npj Computational Materials

---

> ### Author Response · Authors · 2022-11-18
> **Response to Reviewer iZzs**
>
> Dear Reviewer iZzs:
>
> Thank you so much for your time and effort in reviewing! For every concern in your comments, we have provided a detailed response. Would you mind checking them to see if they have successfully addressed your questions and concerns? If you have other questions or you require additional clarification, please let us know soon so that we can provide more responses before the author-reviewer discussion period ends.
>
> Best Regards,
> Authors

---

> ### Author Response · Authors · 2022-11-23
> **We are looking forward to discussing with you**
>
> Dear Reviewer iZzs,
>
> We thank you again for your time and comments. As the time windows are closing, we hope we discuss this with you further to see if our answers have addressed your concerns. If there are any further questions, we can provide further answers to them.
>
> Specifically, we have: (1) clarified our contribution on the dataset side; (2) provided additional experiments on using single-stage sparse regularizers; (3) explained the data representations. We genuinely hope you could check our response and see if we have addressed all the concerns. Thank you very much in advance!
>
> Best Wishes,
>
> Authors

---

> ### Comment · Reviewer_iZzs · 2022-11-28
> **Thanks for the rebuttal**
>
> I thank the authors for the rebuttal. I am satisfied with most of the answers provided for various questions & concerns.
> Thanks for uploading the updated paper. I still have some concerns with baselines. While the provided comparison is helpful, I still have concerns. I think a more extensive analysis might be needed. For example, comparing the approach with a standard few shot setup in Table 1 (like MiniImageNet->CUB) so that you do not have to reproduce & tune baselines and you can directly quote numbers from published papers. Further, more standard few shot baselines for the material science experiments would help a future draft.
>
> The next higher rating is 8. Based on my confidence and the above general concern, I'll retain my rating at 6. I might upgrade my score after discussion with other reviewers.

---

> > ### Author Response · Authors · 2022-11-28
> > **Thank you so much!**
> >
> > Dear Reviewer iZzs,
> >
> > We're so happy you're able to check and acknowledge our response (including our dataset contribution).
> >
> > Regarding your last remaining suggestion, we will start working on more few-shot baselines and will post results here once we obtain updates.
> >
> > Best,

---

### Author Response · Authors · 2022-11-16
**General Responses (Summary of rebuttals)**

We thank all three reviewers for their constructive comments and suggestions, and for unanimously appreciating our motivation, clarity, and strong connection between ML and science. We also thank Reviewer THpP for bringing up the dataset originality issue - we apologize for the confusion caused to non-material science domain researchers, and clarify all below. **In summary, we most certainly claim that the dataset is indeed original, involves substantial effort, and does not concern any integrity issue at all (we do care!),**

We summarize the updates to our manuscript and the additional results we provide during this rebuttal period. Regarding experiments, we have added comparisons against (1) a cross-domain few-shot learning method AFA [r1], (2) a noisy label learning algorithm (which we call Universal Probabilistic Model) [r2], and (3) a Bayesian linear regression model [r3]. These additional baselines have further highlighted the superiority of our method and the novelty of our problem setting. These results are summarized in the table below:

| Method | Testing Accuracy (10% training data) |  Testing MSE (10% training data) |
| :--: | :--: | :--: |
| Bi-RPT (ours) | 66.83% | 0.092 |
| AFA | 57.36% | N/A |
| Universal Probabilistic Model | 54.38% | N/A |
| Bayesian | N/A | 0.100 |


Following Reviewer THpP’s suggestion, we have extended our method to support uncertainty quantification. We calculate the standard deviation in predictions due to model ensembling as the metric of uncertainty. In the table below, we show that an ensemble of sparse models provides more reliable results compared to the pretrain-and-transfer baseline (or dense ensemble). Results are summarized next for ensembles with Bi-RPT and PT in the following format: MSE (uncertainty).

| Alloy | Temperature | Bi-RPT| PT | Bi-RPT-Ensemble | PT-Ensemble |  Experimental (GPa) |
| :---: | :---: | :---: | :---: | ---: | :---: | :--: |
| MoNbTaTi | 293.15 | 1.078 | 1.062 | 1.158 (0.083) | 1.054 (0.011) | 1.210|
| MoNbTaTi | 473.15 | 0.965 | 0.902 | 1.046 (0.087) | 0.908 (0.015) | 0.868 |
| MoNbTaTi | 673.15 | 0.746 | 0.731 | 0.850 (0.085) | 0.740 (0.026)|  0.685 |
| MoNbTaTi | 873.15 | 0.508 | 0.584 | 0.674 (0.103) | 0.604 (0.021) | 0.593 |
| MoNbTaTi | 1273.15 | 0.425 | 0.499 | 0.482 (0.088) | 0.501 (0.018) | 0.539 |
| MoNbTaTiW | 296.15 | 1.268 | 1.068 |  1.268 (0.098) | 1.062 (0.011) | 1.399 |
| MoNbTaTiW | 873.15 | 0.677 | 0.607 | 0.798 (0.102) | 0.624 (0.022) | 0.689 |
| MoNbTaTiW | 1073.15 | 0.618 | 0.523 | 0.681 (0.111) | 0.528 (0.013) |  0.674 |
| MoNbTaTiW | 1273.15 | 0.536 | 0.486 | 0.567 (0.124) | 0.496 (0.017) | 0.620 |
| HfMoNbTaTiZr | 296.15 | 1.527 | 1.132 | 1.392 (0.122)  | 1.142 (0.021)  | 1.515 |
| HfMoNbTaTiZr | 873.15 | 0.861 | 0.685 | 0.864 (0.098) | 0.698 (0.017) | 0.973 |
| HfMoNbTaTiZr | 1073.15 | 0.762 | 0.612 | 0.747  (0.105) | 0.624 (0.022) |  0.791 |
| HfMoNbTaTiZr | 1273.15 | 0.662 | 0.573 | 0.646 (0.134) | 0.587 (0.022) | 0.753 |

The uncertainty estimation of our method is correlated with the prediction error better compared to the PT baseline, indicating that Bi-RPT predicts more reliable results. We have also notice a bonus point: a majority of results are improved after ensembling as the MSE decreases from 0.092 to 0.084.

One frequently asked question is regarding the dataset. We reiterate that the simulation data are calculated by us, following the theoretical computation model from the work of Marseca and Curtin (2020). They are not directly crawled from any existing work and are not trivial to compute (a detailed clarification is provided later). Regarding the real experiment data, both  Borg et al. (2020) and we collected data from published material science articles ( in contrast to some reviewers’ misunderstanding that Borg et al. (2020) generated their own data - as it is a truly infeasible task). We collect our data using a similar workflow to  Borg et al. (2020)  - that is why we cited them. But as opposed to Borg et al. (2020), we focus on a specific subset of alloys R-HEA with BCC structures - another non-trivial curation process requiring domain expertise. We have also separately addressed the reviewers’ questions regarding the dataset part.

We hope our responses below could address all reviewers' concerns. If you should have any further questions, please let us know and we will be happy to address them. We thank all the reviewers again for the time.


[r1] Adversarial Feature Augmentation for Cross-domain Few-shot Classification

[r2] Tackling Instance-Dependent Label Noise via a Universal Probabilistic Model

[r3] Multi-fidelity regression using artificial neural networks: efficient approximation of parameter-dependent output quantities

---

### Author Response · Authors · 2022-11-28
**Public call to AC/SAC and all reviewers: this paper needs special attention and serious discussion**

Dear AC/SAC and all reviewers,

We are eagerly advocating for your attention to this paper since this is not an ordinary "boundary case". This paper received exceptionally **serious accusations** from some of the reviewers, which we, unfortunately, found to be **completely inaccurate and ungrounded**. We made very detailed responses to the raised point.

In particular, **Reviewer THpP** accused us of concerns about our new dataset copyright/integrity which seem to account for the most part of her/his low rating. She/he even flagged our submission as "For Ethics Review: Yes, Legal compliance (e.g., GDPR, copyright, terms of use), Yes, Research integrity issues (e.g., plagiarism, dual submission)".

This accusation, which surprised us a lot, seems to arise from the reviewer being not quite familiar with how material science data was collected, curated, and filtered. In a short answer (more elaborate details are shared in our individual responses):

(1) the simulation data are calculated by us following the theoretical computation model from the work of Marseca and Curtin (2020). They are not directly crawled from any existing work and are not trivial to compute (a detailed clarification is provided later).

(2) Regarding the real experiment data, both Borg et al. (2020) and we collected data from published material science articles ( in contrast to some reviewers’ misunderstanding that Borg et al. (2020) generated their own data - as it is a truly infeasible task). We collect our own data using a similar workflow to Borg et al. (2020) - that is why we cited them. But as opposed to Borg et al. (2020), we focus on a specific subset of alloys R-HEA with BCC structures - another non-trivial curation process requiring domain expertise. We have also separately addressed the reviewers’ questions regarding the dataset part.

The authors’ institution has already completed the legal review and there are no IPO issues at all. The authors are veterans publishing in ML conferences who are familiar with best practices, and their institutions have renowned rigorous publication legal review processes.

We understand some of you might be enjoying holidays, or traveling to NeurIPS. We apologize for bothering you. Yet, due to the very serious nature of the accusation made, and the well-known public record nature of OpenReview forums, **we simply cannot let this seriously wrong accusation regarding our work integrity be left here publicly and permanently**.

We respectfully ask the SAC, AC and reviewers to take responsibility and look at this matter with the notion of academic fairness. As we can all agree on, **making such strong yet inaccurate accusations, with no further response nor revision (after the response has been live online for over 1 week), isn't academically fair or appropriate**.

We thank you all for your time.

Authors,

---

> ### Comment · Reviewer_THpP · 2022-11-28
> **Please refer to the Ethics Committee**
>
> The authors seem to have misunderstood what "Flag For Ethics Review" means. Please note that it is not an accusation, but a recommendation to be further reviewed by the Ethics Committee given potential issues with obtaining data from published papers / simulations to construct a new dataset. Please reach out to the Ethics Committee if this is a concern.

---

> > ### Comment · Area_Chair_WNsL · 2022-12-12
> > **Re**
> >
> > Please note, I do not consider there to be any "serious accusations" here.  Rather an experienced ML reviewer flagged that they are uncertain about how to evaluate whether there are potential concerns with the data used.  The reason for ethics review is to have someone more specialized in the matter take a look.
> >
> > AC

---

> > > ### Author Response · Authors · 2022-12-12
> > > **Yes, we completely agree**
> > >
> > > We totally agree with the AC’s assessment, and are standing by for any required clarification by the ethics review.
> > >
> > > Again, we are very thankful for the reviewer and AC’s time spent on clarifying this matter. Our chosen wording “accusation” was indeed not accurate and doesn’t precisely reflect our awareness either. We apologize for the wording and reiterate our full respect for the reviewer’s expression of concerns.

---

### Decision · Program_Chairs · 2023-01-20

**Decision:**

Reject

**Justification For Why Not Higher Score:**

While ICLR does accept ML for science applications, the authors fail to place their work in the context of the existing literature in few-shot / transfer learning.  The baselines presented are not strong, and so the reviewers struggled to understand the significance of the methodological contribution of the authors and the difficulty of the presented benchmark.

**Justification For Why Not Lower Score:**

N/A

**Metareview: Summary, Strengths And Weaknesses:**

This paper addresses a challenging problem from material science using machine learning, predicting yield strength for high-entropy alloys using few-shot transfer learning.  The authors formulate the problem and present as contributions both a new dataset / benchmark (X-Yield) and a new method (Bi-RPT).   Overall, reviews were mixed with two borderline accepts and one reject (6, 6, 3).  The reviewers all seemed to find the problem compelling and found the work well motivated and well written.  The framing of this as a few-shot transfer learning problem seems well justified and this could be an interesting dataset for methodological researchers to use for development.

Two of the three reviewers found that the empirical evaluation lacked strong baselines.  There are many approaches from the literature that seem stronger than the baselines used.  Herein lies a challenge for reviewers: How can they confidently evaluate a methodological contribution if isn't evaluated on either established benchmarks or compared directly to existing (near) state-of-the-art.  Similarly, evaluating the benchmark is challenging because the lack of established strong baselines makes it hard to evaluate how challenging it is (and place it in the context of existing benchmarks).  Similarly, the reviewers signaled that a discussion of significant related literature was missing as well.  The authors should place their contributions in the context of recent literature in the relevant (sub)field.

One reviewer noted that because the proposed dataset was a combination of data from various sources, i.e. real data from previous literature and simulated, there was the potential to raise ethical concerns.  This reviewer raised an ethics flag for further review.  In personal discussion, the reviewer indicated that this did not lower their review score.  Data from heterogeneous sources can be problematic, so it seems reasonable to review carefully.   The ethics review found that there were no major problems and the clarifications by the authors were sufficient to remove any suspicion of ethics violation.

While ICLR does welcome scientific applications of machine learning, ML for Science, the authors should place this appropriately within the context of the existing literature in ML, both in the discussion and empirically.  Therefore the recommendation is to reject the paper.  It seems that adding stronger baselines and discussion about the state-of-the-art would make this a stronger submission to a future venue.  Note, perhaps a journal such as TMLR would be appropriate since it would allow for some back-and-forth with reviewers.

**Summary Of Ac-Reviewer Meeting:**

I met with one reviewer in person and discussed their concerns about the paper.  In particular, we discussed the ethical concerns regarding the dataset and verified that it was independent of the review score.  I articulated this to the authors (and then acknowledged).  The other concerns were brought up in the back and forth with the authors.